# Avoiding exp(R) scaling in RLHF through Preference-based Exploration

**Mingyu Chen**
Department of Electrical & Computer Engineering
Boston University
Boston, MA 02215
mingyuc@bu.edu

**Yiding Chen**
Department of Computer Science
Cornell University
Ithaca, NY 14850
yc2773@cornell.edu

**Wen Sun**
Department of Computer Science
Cornell University
Ithaca, NY 14850
ws455@cornell.edu

**Xuezhou Zhang**
Faculty of Computing & Data Sciences
Boston University
Boston, MA 02215
xuezhouz@bu.edu

## Abstract

Reinforcement Learning from Human Feedback (RLHF) has emerged as a pivotal technique for large language model (LLM) alignment. This paper studies the setting of online RLHF and focuses on improving its sample efficiency. All existing algorithms for online RLHF, whether doing passive exploration or active exploration, suffer from a sample complexity that scales exponentially with the range of the reward function. This statistical inefficiency hinders their effectiveness in scenarios with heavily skewed preferences, e.g. questions with objectively correct answers. To address this, we introduce *Self-Exploring Preference-Incentive Online Preference Optimization* (SE-POPO), an online RLHF algorithm that for the first time achieves a sample complexity that scales *polynomially* with the reward range, answering an open problem raised by Xie et al. [2024]. Theoretically, we demonstrate that the sample complexity of SE-POPO dominates that of existing exploration algorithms. Empirically, our systematic evaluation confirms that SE-POPO is more sample-efficient than both exploratory and non-exploratory baselines, in two primary application scenarios of RLHF as well as on public benchmarks, marking a significant step forward in RLHF algorithm design.

## 1 Introduction

Reinforcement Learning from Human Feedback (RLHF) has emerged as a pivotal technique in the post-training of Large Language Models (LLMs) [Christiano et al., 2017, Ziegler et al., 2019, Ouyang et al., 2022]. Earlier works on RLHF focus primarily on the offline setting [Ouyang et al., 2022, **?**], where the preference data are pre-collected and fixed prior to the fine-tuning phase. However, in this setting, the quality of alignment is fundamentally limited by the quality of response in the pre-collected preference dataset. To overcome this limitation, recent works attempt to perform RLHF in an online setting. By continually generating and subsequently labeling new samples during training, online RLHF allow the agents to receive feedbacks on out-of-distribution (OOD) responses and achieves improved empirical performance [Dong et al., 2024].

Similar to online reinforcement learning, the most critical challenge in online RLHF is how to balance the *exploration-exploitation trade-off*. In naive online RLHF algorithms [Guo et al., 2024], the

39th Conference on Neural Information Processing Systems (NeurIPS 2025).

exploration is carried out passively, relying solely on the inherent randomness of the LLM policy. Such a passive approach fails to sufficiently explore the prompt-response space even with many samples. More recently, a number of active exploration algorithms have been proposed [Dwaracherla et al., 2024, Xiong et al., 2024a, Xie et al., 2024, Cen et al., 2024, Zhang et al., 2024]. By leveraging optimism-based approaches to encourage the policy to target OOD regions, active exploration has demonstrated superior performance over passive exploration both in theory and in practice. A more comprehensive discussion on related works is deferred to Appendix A.

However, all existing online RLHF algorithms share one common flaw: They remain effective only when the reward is small. In particular, under the Bradley–Terry (BT) model and assuming the reward satisfies $r \in [0, R_{\max}]$, all existing algorithm have a sample complexity in the form of $O(\exp(R_{\max})/\epsilon^2)$, scaling *exponentially* with $R_{\max}$. Intuitively, this issue arises because human feedback in RLHF is given in the form of preferences instead of numerical rewards. Under the BT model, even if there is a significant gap in rewards between two responses, they may behave very similar in their chance of being preferred when pairing with another response that is significantly worse than both. As a result, exponentially many samples are necessary to distinguish the quality of responses based on preference signals. This leads to the open question raised by Xie et al. [2024]:

**Does there exist an online RLHF algorithm that avoids the exponentially dependency on the reward scale?**

In this work, we answer this question in the positive with a new online RLHF algorithm, *Self-Exploring Preference-Incentive Online Preference Optimization* (SE-POPO), that for the first time achieves a sample complexity that scales *polynomially* with the reward scale. Our algorithm is provably sample-efficient, scalable and easy to implement. We summarize our contributions below.

- We introduce a preference-based exploration technique, distinct from the reward-based exploration done in all prior works. Based on this new technique, we design a subroutine algorithm *Preference-Incentive Online Preference Optimization* (POPO), which achieves a preference-based regret that scales polynomially with $R_{\max}$ against a fixed comparator policy.
- Building upon POPO, we propose a self-sampler update technique that effectively prevents the sample complexity from exploding as reward scale increases. Leveraging this idea, we develop our main algorithm SE-POPO, achieving a sample complexity scaling polynomially with $R_{\max}$.
- We perform a comprehensive empirical evaluation of our algorithm across multiple training and testing settings as well as on major public benchmarks. In addition, we perform ablation studies to further understand the effect of the sampler update mechanism in our algorithm. The results show that our algorithm outperforms both exploratory and non-exploratory baselines across all benchmarks with a large margin.

## 2 RLHF Preliminaries

In RLHF, we denote a policy by $\pi$, which generates an answer $y \in \mathcal{Y}$ given a prompt $x \in \mathcal{X}$ according to the conditional probability distribution $\pi(\cdot|x)$. Given two responses $y$ and $y'$ with respect to prompt $x$, we assume a preference oracle, i.e. a human evaluator, will evaluate the quality of two responses and indicate the preferred one. Following prior works, we consider Bradley–Terry model as the preference oracle. The mathematical definition is below.

**Assumption 2.1.** (Bradley–Terry (BT) Model) There exists an underlying reward function $r^\star : \mathcal{X} \times \mathcal{Y} \to \mathbb{R}$ such that for every $x, y, y' \in \mathcal{X} \times \mathcal{Y} \times \mathcal{Y}$,

$$\mathbb{P}^\star(y \succ y'|x) = \frac{\exp(r^\star(x,y))}{\exp(r^\star(x,y)) + \exp(r^\star(x,y'))} = \sigma(r^\star(x,y) - r^\star(x,y')),$$

where $\mathbb{P}^\star(y \succ y'|x)$ represents the probability that $y$ is preferred to $y'$ given $x$ and $\sigma$ represents the sigmoid function. Without loss of generality, we assume that for all $x, y \in \mathcal{X} \times \mathcal{Y}$, we have $r^\star(x,y) \in [0, R_{\max}]$ and $R_{\max} \geq 1$.

**The Two-stage RLHF pipeline:** In the classic two-stage RLHF framework [Christiano et al., 2017, Ouyang et al., 2022], the algorithm assumes access to a dataset $\mathcal{D} = \{x_n, y_n^1, y_n^2, o_t\}_{n=1}^N$, where

$$x_n \sim \rho, \; y_n^1 \sim \pi_{\text{ref}}, \; y_n^2 \sim \pi_{\text{ref}}, \; o_n \sim \text{Ber}\left(\mathbb{P}^\star(y \succ y'|x)\right).$$

Here, $\rho$ denotes the underlying prompt distribution. $\pi_{\text{ref}}$ is a reference language model, which is typically obtained via supervised fine-tuning. $o_n$ is obtained by the preference oracle. For simplicity, we redefine the dataset as $\mathcal{D} = \{x_n, y_n^w, y_n^l\}_{n=1}^N$, where $y_n^w$ and $y_n^l$ are assigned based on the value of $o_n$. Given the dataset, we first estimate the reward function via maximum likelihood estimation, i.e.,

$$\hat{r} = \arg \min_{r \in \mathcal{R}} - \sum_{n=1}^N \log \sigma \left( r(x_n, y_n^w) - r(x_n, y_n^l) \right) =: \arg \min_{r \in \mathcal{R}} \ell(r, \mathcal{D}). \tag{1}$$

With the learned reward function, the objective of RLHF is to fine-tune the policy $\pi$ to maximize the reward. Following prior theoretical works on RLHF, we consider a KL-regularized reward objective, that is,

$$\hat{\pi} = \arg \max_{\pi \in \Pi} \mathbb{E}_{x \sim \rho, y \sim \pi(\cdot|x)} \left[ \hat{r}(x, y) - \beta \log \frac{\pi(y|x)}{\pi_{\text{ref}}(y|x)} \right] =: \arg \max_{\pi \in \Pi} J(\hat{r}, \pi). \tag{2}$$

**The DPO pipeline:** An alternative approach of RLHF is introduced by [**?**], namely Direct Preference Optimization (DPO). The key motivation of DPO is from the closed-form solution of (2), that is, given a reward function $\hat{r}$, the solution $\hat{\pi}$ satisfies

$$\hat{\pi}(y|x) = \frac{\pi_{\text{ref}(y|x)} \exp(\hat{r}(x, y)/\beta)}{Z(r, x)}, \ \forall x, y \in \mathcal{X} \times \mathcal{Y} \tag{3}$$

where $Z(r, x) = \sum_y \pi_{\text{ref}(y|x)} \exp(\hat{r}(y|x)/\beta)$ is a partition function independent of $y$. The closed form solution allows us to represent the reward by $\hat{\pi}$

$$\hat{r}(x, y) - \hat{r}(x, y') = \beta \log \frac{\hat{\pi}(y|x)}{\pi_{\text{ref}}(y|x)} - \beta \log \frac{\hat{\pi}(y'|x)}{\pi_{\text{ref}}(y'|x)} \tag{4}$$

for every $\forall (x, y, y') \in \mathcal{X} \times \mathcal{Y} \times \mathcal{Y}$. By substituting (4) into (1), DPO bypasses the need for explicitly learning the reward function. Instead, it optimizes the policy directly with objective

$$\hat{\pi} = \arg \min_{\pi \in \Pi} - \sum_{n=1}^N \log \sigma \left( \beta \log \frac{\pi(y_n^w|x_n)}{\pi_{\text{ref}}(y_n^w|x_n)} - \beta \log \frac{\pi(y_n^l|x_n)}{\pi_{\text{ref}}(y_n^l|x_n)} \right) =: \arg \min_{\pi \in \Pi} \ell(\pi, \mathcal{D}). \tag{5}$$

**Performance metric** The performance of a learned policy $\hat{\pi}$ is measured by the suboptimal gap

$$\text{SubOpt}(\hat{\pi}) = \mathbb{E}_{x \sim \rho, y \sim \pi^\star(\cdot|x), y' \sim \hat{\pi}(\cdot|x)}[r^\star(x, y) - r^\star(x, y')],$$

where $\pi^\star = \arg \max_{\pi \in \Pi} \mathbb{E}_{x \sim \rho, y \sim \pi^\star(\cdot|x)}[r^\star(x, y)]$ denotes the optimal policy. Our goal is to propose a sample-efficient and also implementation-friendly algorithm to learn a policy $\hat{\pi} \in \Pi$ such that $\text{SubOpt}(\hat{\pi}) \leq \epsilon$ for some small $\epsilon > 0$.

**Online Feedback and Exploration** In early RLHF studies, the preference dataset $\mathcal{D}$ is typically assumed to be given. Although such offline RLHF has been highly successful in aligning language models, it is inherently limited by the quality of the preference data and $\pi_{\text{ref}}$. To overcome these limitations, RLHF with online feedback is proposed [Guo et al., 2024]. In the online framework, the dataset is constructed with human feedbacks on the responses generated from the language model on the fly. Formally, online RLHF proceeds in $T$ rounds with each round as follows:

1. The agent computes $\pi_t$ using current data $\mathcal{D}_t$ and samples $x_t \sim \rho, y_t^1 \sim \pi_t(\cdot|x), y_t^2 \sim \pi_t(\cdot|x)$.
2. Human labels responses $(x_t, y_t^1, y_t^2) \rightarrow (x_t, y_t^w, y_t^l)$. Update $\mathcal{D}_{t+1} = \mathcal{D}_t \cup \{(x_t, y_t^w, y_t^l)\}$.

Although numerous empirical studies have demonstrated the benefits of online RLHF, the theoretical foundation has been missing. The main reason is that existing methods rely on *passive exploration* to collect data, i.e. the responses are sampled directly from the policy $\pi_t$ relying purely on the randomness of $\pi_t$ for exploration. Motivated by this, recent works [Cen et al., 2024, Xie et al., 2024, Zhang et al., 2024] start to incorporate the optimism principle into RLHF, which encourages explicitly exploration in the policy $\pi_t$. Although their implementations differ, the essence of their algorithms is to replace the MLE objectives (1) and (2) in vanilla RLHF with

$$r_{t+1} = \arg \max_{r \in \mathcal{R}} \left\{ -\ell(r, \mathcal{D}_t) + \alpha J(r, \pi(r)) \right\}, \ \text{s.t.} \ \pi(r) = \arg \max_{\pi \in \Pi} J(r, \pi) \tag{6}$$

where $\alpha \max_{\pi \in \Pi} J(r, \pi)$ is a **reward-based exploration bonus** that encourages exploration. Such a bonus leads to an overestimation of rewards with high uncertainty, thereby incentivizing policy to explore uncertain responses. As shown by Cen et al. [2024], Xie et al. [2024], this design offers a practical and provably sample-efficient online exploration algorithm for RLHF with general function approximation.

## 3 Preference-based Exploration

Although existing algorithms based on (6) obtain theoretical sample efficiency guarantees, there is a significant gap between their bounds and what could be achieved under the standard MDP framework. In particular, the best known sample complexity bound takes the form of $O(\exp(R_{\max})/\epsilon^2)$, which scales *exponentially* with the reward scale $R_{\max}$. This makes existing guarantees quite subtle, as the bound quickly becomes vacuous as soon as $R_{\max}$ is moderately large. In practical LLM applications, it is common that one response can strictly dominate another, i.e., $\mathbb{P}^\star(y \succ y'|x) \to 1$. Under the BT model (Asm. 2.1), this implies a very large $R_{\max}$. Authors of prior works have admitted that this is a significant drawback of these results and in fact conjectured that the exponential dependency might be unavoidable [Xie et al., 2024]. In this paper, we resolve this conjecture in the negative by presenting the first algorithm that avoids such exponential dependency on the reward scale. In what follows, we start by discussing the cause of exponential dependency on $R_{\max}$ and why it's a real limitation of the algorithms rather than merely an artifact of the analysis. After that, we will present our technique that solves it.

### 3.1 The cause of $\exp(\mathbf{R_{\max}})$ scaling

Using online-to-batch technique, the sample complexity of an online algorithm can be derived from its regret, which is defined by $\sum_{t=1}^{T} \text{SubOpt}(\pi_t)$. In the standard analysis of optimism online RLHF, the regret can be bounded by the sum of reward uncertainty, i.e., $\sum_{t=1}^{T} \mathbb{E}_{x\sim\rho, y\sim\pi_t(\cdot|x)}[|r_t(x,y) - r^\star(x,y)|]$, where $r_t$ is the induced reward function from $\pi_t$ as in DPO. To bound the reward uncertainty, prior works reduce it to the preference uncertainty, i.e., $\sum_{t=1}^{T} \mathbb{E}_{x\sim\rho, y\sim\pi_t(\cdot|x), y'\sim\pi_t(\cdot|x)}[|\mathbb{P}_t(y \succ y'|x) - \mathbb{P}^\star(y \succ y'|x)|]$, as the preference uncertainty can be effectively bounded using concentration inequalities. Unfortunately, this reduction is not a free lunch: due to the presence of sigmoid function in Bradley–Terry Model, for some $x, y, y'$, there is

$$|r_t(x,y) - r^\star(x,y)| \approx \frac{|\mathbb{P}_t(y \succ y'|x) - \mathbb{P}^\star(y \succ y'|x)|}{\nabla\sigma(r^\star(x,y) - r^\star(x,y'))} \tag{7}$$

Therefore, the reward uncertainty could be of order $1/\nabla\sigma(R_{\max}) \approx \mathcal{O}(\exp(R_{\max}))$ times the preference uncertainty, in the worst case where the reward gap between the two responses $y$ and $y'$ is large. Similar issues have also been discovered in logistic bandits [Faury et al., 2020]. This explains where $\exp(R_{\max})$ comes from in the theoretical analysis of existing works and highlights the key question in algorithm design: **How should we sample the responses $y$ and $y'$ in online RLHF?** A number of prior works [Xiong et al., 2024a, Dong et al., 2024, Shi et al., 2024] use $\pi_t$ to sample $y_t^1$ and use $\pi_{\text{ref}}$, or a policy distinct from $\pi_t$, to sample $y_t^2$. This destines to perform poorly due to (7). In general, sampling $y_t^2$ using an underperformed policy, such as $\pi_{\text{ref}}$ implies that the reward gap $r^\star(x_t, y_t^1) - r^\star(x_t, y_t^2)$ would be relatively large, causing $y_t^1$ to be consistently favored, even if $y_t^1$ itself is suboptimal. As a result, such algorithms will struggle to learn the optimal response, as such a comparison provides very little information on how to improve based on the current best policy $\pi_t$.

### 3.2 Algorithm Design

Given the above intuition, we propose *Preference-Incentive Online Preference Optimization with Self-updated Sampler* (`SE-POPO`), which for the first time enjoys a sample complexity bound that scales **polynomially with $R_{\max}$**. Conceptually, `SE-POPO` differs from prior algorithms in two main aspects: 1) it uses a preference-based exploration bonus instead of a reward-based bonus to explore more efficiently, and 2) it updates the second sampler at intervals instead of fixing it as $\pi_{\text{ref}}$, bypassing the design flaw discussed above. The pseudocode of the algorithms is presented in Algorithm 1 and 2.

`SE-POPO` operates over $K$ intervals. In each interval, `SE-POPO` selects a fixed sampler $\pi_{\text{sam}}$ to generate the second response and runs the subroutine `POPO` for $T$ iterations. The output of `POPO` is used as the sampler for the next interval, and the output from the last interval serves as the output of `SE-POPO`. Let us now present the subroutine `POPO`. As illustrated in Algorithm 2, `POPO` shares a similar structure with existing optimism RLHF algorithms [Xie et al., 2024, Zhang et al., 2024, Cen et al., 2024]. However, unlike prior designs that are tailored towards bounding the **reward-based regret**, i.e. $\sum_{t=1}^{T} \text{SubOpt}(\pi_t)$, `POPO` takes an indirect approach and instead optimize the **preference-based**

**Algorithm 1** SE-POPO: Self-Exploring Preference-Incentive Online Preference Optimization
***
**Input:** Reference policy $\pi_{\text{ref}}$, Policy set $\Pi$, Iterations $T$, Intervals $K$
Initialize $\pi_{\text{sam}}^1 \leftarrow \pi_{\text{ref}}$.
**for** $k = 1, \ldots, K-1$ **do**
    Update the sampler $\pi_{\text{sam}}^{k+1} \leftarrow \texttt{POPO}(\pi_{\text{ref}}, \pi_{\text{sam}}^k, \Pi, T)$.
**end for**
Return policy $\bar{\pi} = \texttt{POPO}(\pi_{\text{ref}}, \pi_{\text{sam}}^K, \Pi, T)$.
***

**regret** over a fixed sampler $\pi_{\text{sam}}$:

$$\text{Reg}_{\text{pref}}(\pi_{\text{sam}}, T) := \sum_{t=1}^{T} \mathop{\mathbb{E}}_{x \sim \rho, (y^\star, y, y') \sim \pi^\star \otimes \pi_t \otimes \pi_{\text{sam}}(\cdot|x)} \left[ \mathbb{P}^\star(y^\star \succ y'|x) - \mathbb{P}^\star(y \succ y'|x) \right]. \quad (8)$$

To achieve this, POPO optimizes the following objective function instead of (6):

$$r_{t+1} = \arg\max_{r \in \mathcal{R}} \left\{ -\ell(r, \mathcal{D}_t) + \alpha G(r, \pi(r)) I(r, \mathcal{D}_t) \right\}, \text{ s.t. } \pi(r) = \arg\max_{\pi \in \Pi} J(r, \pi) \quad (9)$$

Here, $G$ is the expected preference rate of $\pi$ over $\pi_{\text{sam}}$

$$G(r, \pi) = \mathbb{E}_{x \sim \rho, y \sim \pi(\cdot|x), y' \sim \pi_{\text{sam}}(\cdot|x)} \left[ \mathbb{P}_r \left( y \succ y'|x \right) \right],$$

where $\mathbb{P}_r$ denotes the preference oracle parameterized by reward $r$. $I(\mathcal{D}_t)$ is an indicator function

$$I(r, \mathcal{D}_t) = \mathbb{1} \left\{ \ell(r, \mathcal{D}_t) - \ell(\bar{r}, \mathcal{D}_t) \leq \gamma \right\},$$

where $\bar{r} = \arg\min_{r \in \mathcal{R}} \ell(r, \mathcal{D}_t)$ represents the MLE-based reward estimator. In brief, POPO applies a truncated preference-based exploration bonus on the reward learning objective. This design ensures that optimistic exploration is conducted directly with respect to preferences rather than rewards, while also constraining the exploration to regions near the current MLE estimator, thereby mitigating the risk of over-exploration.

Assuming that POPO achieves low preference-based regret, let's now look at how POPO with self-updated samplers eliminates the exponential dependence on $R_{\text{max}}$ in the reward-based regret. The key observation is a novel *Preference-to-Reward* reduction lemma as follows.

**Lemma 3.1.** *(Preference-to-Reward reduction) Given any prompt $x \in \mathcal{X}$, let $y^\star$ denotes the optimal response $y^\star = \arg\max_{y \in \mathcal{Y}} r^\star(x, y)$. For every $(y, y') \in \mathcal{Y} \times \mathcal{Y}$, there is $\mathbb{1}\{r^\star(x, y) - r^\star(x, y') \leq 1\}[r^\star(x, y^\star) - r^\star(x, y)] \leq 20R_{max} [\mathbb{P}^\star(y^\star \succ y'|x) - \mathbb{P}^\star(y \succ y'|x)].$*

The proof of Lemma 3.1 is deferred to the appendix. Intuitively, Lemma 3.1 tells us that the exponential blow-up in preference-to-reward reduction only occurs when $r^\star(x, y) - r^\star(x, y')$ is large. Assuming $y' \sim \pi_{\text{sam}}(\cdot|x)$. If $\pi_{\text{sam}}$ is "good enough" such that $r^\star(x, y) - r^\star(x, y') \leq 1$ holds for all $x$, we can easily bound the reward-based regret by $\text{Reg}_r(T) \leq \mathcal{O}(R_{\text{max}})\text{Reg}_{\text{pref}}(\pi_{\text{sam}}, T)$, and thus get rid of the exponential dependence on $R_{\text{max}}$. So how do we find a good enough sampler $\pi_{\text{sam}}$? An intuitive idea is to first run POPO to find a suboptimal policy, then use this policy as $\pi_{\text{sam}}$ and rerun POPO. However, notice that finding a good enough policy by running POPO from scratch would still requires $\mathcal{O}(\exp(R_{\text{max}}))$ iterations, as we would have been using $\pi_{\text{ref}}$ as the sampler, and $\pi_{\text{ref}}$ might be $O(R_{\text{max}})$ worse than $\pi^*$. The trick, as shown in Algorithm 1, is to repeat the POPO subroutine for many times and gradually improve $\pi_{\text{sam}}$. The main observation is that even if the sampler performs poorly, POPO's output policy can still achieve a reward higher by a constant amount compared to the sampler. For instance, consider $x, y^\star, y'$ such that $r^\star(x, y^\star) - r^\star(x, y')$ is large. If we use $y'$ as the second response, after $T$ iterations, we can find a $y$ such that $P^\star(y \succ y'|x) \geq P^\star(y^\star \succ y'|x) - \tilde{\mathcal{O}}(1/\sqrt{T})$ by the preference-based regret (8). Since $r^\star(x, y^\star) - r^\star(x, y')$ is large, $P^\star(y^\star \succ y'|x)$ will be close to 1, resulting in $P^\star(y \succ y'|x)$ being significantly greater than $1/2$, which implies that there is a constant improvement between $r^\star(x, y)$ and $r^\star(x, y')$. Therefore, by repeating POPO $K = \mathcal{O}(R_{\text{max}})$ intervals, the sampler will finally become sufficiently effective.

### 3.3 Implementation-friendly Objective

Similar to that of vanilla two-stage RLHF, (9) is a bilevel optimization involving both reward and policy, and is challenging to solve in practice. Fortunately, $\pi(r)$ remains to be the solution to the

---

**Algorithm 2** POPO: Preference-Incentive Online Preference Optimization

---

**Input:** Reference policy $\pi_{\text{ref}}$, Sampler $\pi_{\text{sam}}$, Policy set $\Pi$, Iterations $T$
Initialize $\pi_1 = \pi_{\text{ref}}$.
**for** $t = 1, \ldots, T$ **do**
    Generate data $x_1 \sim \rho, y_t^1 \sim \pi_t(\cdot|x), y_t^2 \sim \pi_{\text{sam}}(\cdot|x)$.
    Label the two responses: $(x_t, y_t^1, y_t^2) \rightarrow (x_t, y_t^w, y_t^l)$.
    Optimize objective (10). Get $\pi_{t+1}$.
**end for**
Return policy $\bar{\pi} = \text{Uniform}(\pi_1, \ldots, \pi_t)$.

---

KL-regularized reward optimization objective, therefore (4) continues to hold. By substituting (4) into (9), similar to what is done in `DPO`, we can bypass the reward model and directly optimize the policy. Therefore, the objective can be rewritten as

$$\pi_{t+1} = \arg\max_{\pi \in \Pi} \left\{ -\ell(\pi, \mathcal{D}_t) + \alpha G(\pi) I(\pi, \mathcal{D}_t) \right\}, \tag{10}$$

where $\ell(\pi, \mathcal{D}_t)$ is the DPO loss as in (5). $G(x)$ is the exploration bonus defined by

$$G(\pi) = \mathbb{E}_{x \sim \rho, y \sim \pi(\cdot|x), y' \sim \pi_{\text{sam}}(\cdot|x)} \left[ \sigma\left( \beta \log \frac{\pi(y|x)}{\pi_{\text{ref}}(y|x)} - \beta \log \frac{\pi(y'|x)}{\pi_{\text{ref}}(y'|x)} \right) \right],$$

and $I(\pi, \mathcal{D}_t) = \mathbb{1}\left\{ \ell(\pi, \mathcal{D}_t) - \ell(\bar{\pi}, \mathcal{D}_t) \leq \gamma \right\}$ with $\bar{\pi} = \arg\min_{\pi \in \Pi} \ell(\pi, \mathcal{D}_t)$. In addition, evaluating $I(r, \mathcal{D}_t)$ requires pre-computing the MLE estimator $\bar{\pi}$ first, which doubles the computation cost. In our experiments, we find that the truncation $I(r, \mathcal{D}_t)$ is rarely active and can therefore be omitted. These steps result in the implementation-friendly objective below

$$\pi_{t+1} = \arg\max_{\pi \in \Pi} \sum_{s=1}^{t} \log \sigma\left( \beta \log \frac{\pi(y_s^w|x_s)}{\pi_{\text{ref}}(y_s^w|x_s)} - \beta \log \frac{\pi(y_s^l|x_s)}{\pi_{\text{ref}}(y_s^l|x_s)} \right)$$
$$+ \alpha \mathbb{E}_{x \sim \rho, y \sim \pi(\cdot|x), y' \sim \pi_{\text{sam}}(\cdot|x)} \left[ \sigma\left( \beta \log \frac{\pi(y|x)}{\pi_{\text{ref}}(y|x)} - \beta \log \frac{\pi(y'|x)}{\pi_{\text{ref}}(y'|x)} \right) \right]. \tag{11}$$

On paper, (11) can already be implemented efficiently into existing online DPO pipeline Guo et al. [2024] with a one-line change of the code. However, one challenge we encounter when implementing (11) is that calculating the gradient of the objective function requires sampling new responses $y \sim \pi(\cdot|x)$. While such sampling is techniquely feasible, we empirically found that this on-policy sampling step is extremely slow in language model finetuning due to the lack of efficient LLM online inference libraries. To bypass this issue, we decide to prune the first term within the bonus all together, resulting in the following objective:

$$\pi_{t+1} = \arg\max_{\pi \in \Pi} \sum_{s=1}^{t} \log \sigma\left( \beta \log \frac{\pi(y_s^w|x_s)}{\pi_{\text{ref}}(y_s^w|x_s)} - \beta \log \frac{\pi(y_s^l|x_s)}{\pi_{\text{ref}}(y_s^l|x_s)} \right)$$
$$+ \alpha \mathbb{E}_{\substack{x \sim \rho \\ y' \sim \pi_{\text{sam}}(\cdot|x)}} \left[ \sigma\left( -\beta \log \frac{\pi(y'|x)}{\pi_{\text{ref}}(y'|x)} \right) \right]. \tag{12}$$

Surprisingly, objective (12) still yields in a sample-efficient algorithm in theory. We defer further discussion on (12) to Appendix B and now move on to presenting our main theoretical results.

### 3.4 Theoretical Guarantees

Let the regularization parameter $\beta > 0$ be fixed. We start by a *reward realizability* assumption, which states that the reward class used in `SE-POPO` is sufficiently expressive.

**Assumption 3.2.** (Reward realizability) There exists a set of reward functions $\mathcal{R}$ satisfying $r^\star \in \mathcal{R}$.

Given Assumption 3.2, we define $\mathcal{P}$ as the set of preference model induced by $\mathcal{R}$, and define $\Pi$ as the optimal policies induced by $\mathcal{R}$ under KL-regularized reward objective (2). Notice that $|\mathcal{P}| = |\mathcal{R}| = |\Pi|$ by definition. For ease of understanding, we will present our main theorems under the linear reward model setting and defer results for general function approximation to Appendix J.

**Assumption 3.3.** (Linear reward oracle) Every reward $r \in \mathcal{R}$ can be parameterized by

$$r_\theta(x, y) = \langle \phi(x, y), \theta \rangle, \ \forall (x, y) \in \mathcal{X} \times \mathcal{Y},$$

where $\phi(x, y) : \mathcal{X} \times \mathcal{Y} \to \mathbb{R}^d$ is a fixed feature mapping and $\theta \in \mathbb{R}^d$ is the parameter. Without loss of generality, we define $\phi(x, y, y') = \phi(x, y) - \phi(x, y')$. We further assume that $|\phi(x, y, y')| \leq 1$ for all $x, y, y'$ and $\|\theta\|_2 \leq R_{\max}$.

The following is the preference-based regret bound for `POPO`.

**Theorem 3.4.** *Given Assumption 3.2 and 3.3, setting $\alpha = \sqrt{\frac{d \log T/d}{R_{max} T \log |\mathcal{R}|/\delta}}$ and $\gamma = 2 \log \frac{|\mathcal{R}|}{\delta}$, with probability $1 - 2\delta$, `POPO` output a policy $\bar{\pi}$ such that*

$$\mathbb{E}_{\substack{x \sim \rho \\ (y^\star, y, y') \sim \pi^\star \otimes \bar{\pi} \otimes \pi_{sam}(\cdot|x)}} \left[ \mathbb{P}^\star(y^\star \succ y'|x) - \mathbb{P}^\star(y \succ y'|x) \right]$$

$$\leq \mathcal{O}\left( \sqrt{\frac{dR_{max} \log \frac{TR_{max}}{d} \log \frac{|\mathcal{R}|}{\delta}}{T}} + \frac{d \exp(R_{max})}{T} \log \frac{TR_{max}}{d} \log \frac{|\mathcal{R}|}{\delta} + \beta C_{KL} \right)$$

*where $C_{KL} = \mathbb{E}_{x \sim \rho} \left[ \mathbb{D}_{KL}(\pi^\star(\cdot|x) || \pi_{ref}(\cdot|x)) \right]$* [1]

Theorem 3.4 established a clean $\tilde{\mathcal{O}}(\sqrt{dT})$ bound on the preference-based regret. This implies, for example, if one were to train against a strong baseline $\pi_{\text{sam}}$, e.g. GPT-4o, `POPO` would achieve a winrate against GPT-4o similar to that of the optimal policy with a fast rate of convergence. Of course, in practice, we may not have such strong baselines at our disposal. `SE-POPO` is designed to achieve a similar performance even without such baselines, by iteratively updating its $\pi_{\text{sam}}$. Our main theorem is presented as follows.

**Theorem 3.5.** *Assuming $C_{KL}$ is well-bounded. Setting $K = \lceil R_{max} \rceil$, with probability $1 - \delta$, `SE-POPO` output a policy $\bar{\pi}$ such that*

$$\mathbb{E}_{\substack{x \sim \rho \\ (y^\star, y) \sim \pi^\star \otimes \bar{\pi}(\cdot|x)}} \left[ r(x, y^\star) - r(x, y) \right] \leq \tilde{\mathcal{O}}\left( \sqrt{\frac{dR_{max}^8 \log \frac{|\mathcal{R}|}{\delta}}{N}} + \beta R_{max}^3 C_{KL} \right).$$

*Specifically, with $\beta = o(1/\sqrt{T})$, `SE-POPO` outputs $\epsilon$-optimal policy with $\tilde{\mathcal{O}}\left( \frac{dR_{max}^8 \log \frac{|\mathcal{R}|}{\delta}}{\epsilon^2} \right)$ samples.*

**Remark 3.6.** Theorem 3.5 offers a significant improvement over all prior sample complexity bounds for RLHF algorithms under the BT-model, being the first sample complexity bound that scales polynomially with $R_{\max}$. Compared to prior works on online RLHF [Das et al., 2024b, Rosset et al., 2024, Xie et al., 2024, Zhang et al., 2024, Cen et al., 2024], Theorem 3.5 retains the same dependencies on the coverage parameter $d$ and precision $\epsilon$, while successfully eliminating the exponential dependence on $R_{\max}$ and $1/\beta$. Furthermore, in Appendix J, we demonstrate that the theoretical results of `POPO` and `SE-POPO` can be generalized beyond linear preference oracle using a general complexity measure proposed in [Zhong et al., 2022], extending our theoretical results to the general function approximation setting.

## 4 Experiments

In this section, we provide a comprehensive empirical evaluation of `SE-POPO` in LLM alignment tasks. There are two primary use cases for LLM alignments in real practices:

1. **Domain-specific alignment**: This is where the goal is to fine-tune LLMs for a specific type of task, e.g. fashion design.
2. **Generalist algnment**: This is where the goal is to train a general-purpose question answering AI that could answer a wide variety of questions. This is for instance what GPTs are designed for.

---

[1] While Theorem 3.4 eliminates the exponential dependence in the leading sample complexity term, there still exists an $\exp(R_{max})$ burn in cost in the lower order term. This is consistent with known lower bounds in logistic bandit literature [Faury et al., 2020] and is believed to be not improvable.

| Model | IID Data | | Alpaca Data | | AE2 LC | MT-Bench | Avg. Len. (in AE2) |
|---|---|---|---|---|---|---|---|
| | WR | AvgR | WR | AvgR | | | |
| Llama-3-8B-SFT | - | -5.56 | - | -7.94 | 10.20 | 7.69 | 1182 |
| DPO-iter1 | 62.4 | -4.50 | 78.1 | -6.02 | - | - | 1645 |
| DPO-iter2 | 66.6 | -3.59 | 87.1 | -3.34 | - | - | 2045 |
| DPO-iter3 | 72.4 | -2.33 | 91.3 | -0.02 | 36.10 | 8.28 | 2257 |
| XPO-iter1 | 62.6 | -4.40 | 78.3 | -5.79 | - | - | 1674 |
| XPO-iter2 | 67.3 | -3.28 | 88.0 | -2.60 | - | - | 2200 |
| XPO-iter3 | 73.0 | -2.09 | 91.8 | 0.60 | 38.23 | 8.21 | 2346 |
| SE-POPO-iter1 | 62.5 | -4.32 | 80.0 | -5.68 | - | - | 1797 |
| SE-POPO-iter2 | 68.2 | -3.15 | 89.1 | -2.45 | - | - | 2302 |
| SE-POPO-iter3 | **73.3** | **-2.03** | **92.4** | **0.61** | **40.12** | **8.39** | 2358 |
| Llama-3-8B-Instruct | 48.4 | -6.77 | 87.0 | -3.42 | 22.92 | 8.16 | 1899 |
| Llama-3-405B-Instruct | - | - | - | - | 39.30 | - | 1988 |

Table 1: Performance comparison across multiple chat benchmarks.

Importantly, in both use cases, the preference feedback during both training and evaluation would have been provided by **the same oracle**, e.g. human evaluators. In other words, there should not be any distribution shift in the underlying preference model between training and testing. What distinguishes the two use cases is the prompt distribution during training and deployment. For use case 1, the prompts should come from the same domain during both training and deployment, i.e. no distribution shift in the prompt distribution. For use case 2, the prompt distribution between training and testing could be different.

Motivated by the real use cases discussed above, we present three sets of experiments. For all experiments, our implementation build upon the iterative DPO codebase from [Dong et al., 2024], and we use the 3-iteration online RLHF framework following the setting in [Xie et al., 2024]. Across all three experiments, we use `Llama-3-8B-SFT` as the base model [2], `RLHFlow-ultrafeedback` dataset as the training prompt sets, and `GRM-Llama3-8B-rewardmodel-ft` as the training preference model. More details about the experiment setup are deferred to Appendix K.1. The results from the three sets of experiments are shown as three columns in Table 1:

- **"IID data"** refers to the setting where the models are evaluated on a held-out test prompt set that are drawn from the same distribution as the training prompt set, and the responses are evaluated by the same preference model used during training. This is to simulate Use Case #1.
- **"Alpaca data"** refers to the setting where the models are evaluated on the AlpacaEval 2.0 dataset, but the responses are still evaluated by the same preference model used during training. This is to simulate Use Case #2.
- **Public benchmarks**: Finally, we also evaluate our algorithm on public benchmarks including AlpacaEval 2.0 and MT-bench shown in Table 1 as well as the academic benchmarks that are deferred to Table 5 in the appendix. These public benchmarks all have one common characteristic: the training and evaluation preference models are different, usually with GPT-4o as the evaluation oracle during testing. As discussed above, such a distribution shift in the preference model between training and testing rarely happen in practice. Thus, we believe that **the performances on such benchmarks offer little insight on how well an RLHF algorithm works in practice**. Nevertheless, we include them for completeness due to their wide adoption in prior RLHF research.

**Baselines:** We compare against two baseline algorithms: iterative DPO [Dong et al., 2024], which is the state-of-the-art passive exploration algorithm and XPO [Xie et al., 2024] which is the state-of-the-art active exploration algorithm. Importantly, here we use the practical implementation of XPO, where both responses are drawn from the previous policy $\pi_t$, rather than from $\pi_t$ and the reference policy $\pi_{\text{ref}}$ [3]. Empirically, this practical variant of XPO substantially outperforms its theoretical counterpart described in the original paper. However, it is important to note that this modification

---

[2]To further verify the robustness of `SE-POPO`, we additionally evaluate with another base model `Qwen2.5-7B-Instruct`, where the results are deferred to Appendix K.2

[3]For completeness, we also test the the theoretical XPO and defer the results to Appendix K.4

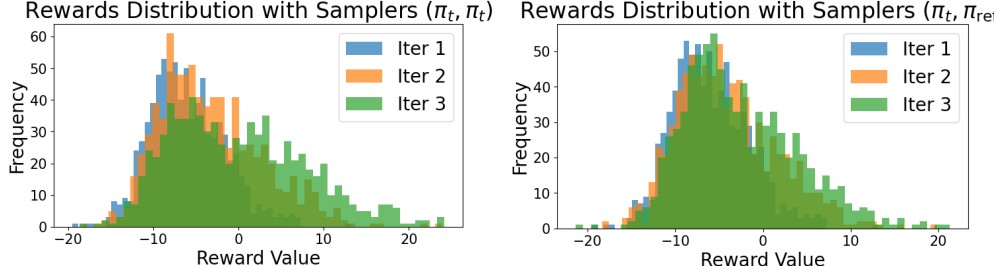

Figure 1: Rewards Distribution with Different Samplers.

| Model | WR | AvgR |
|-------|-----|------|
| $(\pi_t, \pi_t)$-iter2 | 87.0 | -3.35 |
| $(\pi_t, \pi_t)$-iter3 | 91.2 | -0.02 |
| $(\pi_t, \pi_{\text{ref}})$-iter2 | 86.8 | -4.09 |
| $(\pi_t, \pi_{\text{ref}})$-iter3 | 89.4 | -2.63 |

Table 2: Average Reward and Win Rate Comparison.

| Model | Gap Mean | Gap Max | Ratio (Gap > 3) |
|-------|----------|---------|-----------------|
| $(\pi_t, \pi_t)$-iter2 | 2.33 | 23.6 | 0.28 |
| $(\pi_t, \pi_t)$-iter3 | 2.90 | 26.0 | 0.36 |
| $(\pi_t, \pi_{\text{ref}})$-iter2 | 3.84 | 28.4 | 0.49 |
| $(\pi_t, \pi_{\text{ref}})$-iter3 | 7.14 | 33.9 | 0.71 |

Table 3: Reward Gap Statistics.

is **not supported** by the original theoretical analysis. In particular, as shown by Cen et al. [2024], when both responses are drawn from $\pi_t$, the sample complexity scales as $\mathcal{O}(\exp(1/\beta))$, leading to substantially weaker theoretical guarantees. By contrast, our algorithm adheres strictly to its theoretical formulation, which we view as crucial for the principled development of LLM alignment algorithms.

**Results:** As can been seen in Table 1, `SE-POPO` outperforms both `DPO` and `XPO` across all experiment setups. Moreover, on the public benchmarks, `SE-POPO` achieves better performance compared to the industry-level 8B model (Llama-3-8B-Instruct) and comparable performance to model with two orders of magnitude more parameters (Llama-3-405B-Instruct). Beyond instruction-following benchmarks, we also evaluate `SE-POPO` and the baselines on a suite of academic benchmarks, to demonstrate that our improvements in chat capabilities do not come at an additional expense of reasoning ability compared to other baselines. The results are deferred to Appendix K.3. Across the 9 academic tasks evaluated, our algorithm performs best in 4, while `DPO` leads in 3 and `XPO` in 2. These evaluation results resoundingly support the effectiveness of our algorithm.

**Slight length exploitation in `XPO` and `SE-POPO`:** It is worth noting that the length of the responses generated with models trained by `XPO` and `SE-POPO` are slightly longer compared to `DPO`. This makes sense in theory, considering that the exploration term in both `XPO` loss and (12) encourages minimizing $\log \frac{\pi(y'|x)}{\pi_{\text{ref}}(y'|x)}$, which inherently incentives models to generate longer responses. We speculate that using objective (11) can mitigate this exploitation, as the on-policy term $\log \frac{\pi(y|x)}{\pi_{\text{ref}}(y|x)}$ in (11) will encourage $\pi$ to generate shorter responses, thereby counteracting the effect incurred by $\log \frac{\pi(y'|x)}{\pi_{\text{ref}}(y'|x)}$. Unfortunately, we cannot implement the version of `SE-POPO` with objective (11) and have to defer a more comprehensive study of this phenomenon to future work.

**Ablation study on the impact of sampler $\pi_{\text{sam}}$:** We conduct an ablation study to better understand the impact of samplers. We use iterative `DPO` as the base algorithm and consider two sampling subroutines:

1. both responses are sampled by the policy of the previous iteration, i.e., $x \sim \rho, (y^1, y^2) \sim \pi_t(\cdot|x)$;
2. one response is sampled from the previous iteration's policy and one from the initial policy, i.e., $x \sim \rho, y_1 \sim \pi_t(\cdot|x), y^2 \sim \pi_{\text{ref}}(\cdot|x)$.

As shown in Table 2, we study two metrics: 1). the reward corresponding to the responses produced by the models, 2). the win rate with respect to the base model $\pi_{\text{ref}}$. Notice that for both iteration 2 and iteration 3, the difference in win rate between the two sampler settings is relatively small, whereas the discrepancy in average reward is substantial. In addition, we plot the reward distribution of the model

outputs, as illustrated in Figures 1. For samplers $(\pi_t, \pi_{\text{ref}})$, the reward distribution remains relatively unchanged between iteration 2 and 3. In contrast, samplers $(\pi_t, \pi_t)$ demonstrates a more pronounced change in the reward distribution. These results are consistent with our theoretical intuition in Section 3.1: collecting data by $(\pi_t, \pi_{\text{ref}})$ can result in $\pi_t$ consistently winning, thereby limiting its capacity to acquire new information. Consequently, the models can only learn a policy that is sufficiently better than $\pi_{\text{ref}}$ (with 86% and 89% win rate), but fail to improve any further.

As shown in Table 3, we investigate the reward gap between paired responses $(y_1, y_2)$ under the two sampling schemes. We report three metrics: the mean reward gap, the maximum reward gap, and the proportion of large gaps (i.e., Gap $> 3$). Compared to sampler $(\pi_t, \pi_{\text{ref}})$, $(\pi_t, \pi_t)$ consistently produces smaller gaps across all metrics. In particular, at iteration 3, $(\pi_t, \pi_{\text{ref}})$ exhibits a substantial fraction of large gaps (ratio = 0.71), indicating that the effect of $\exp(R_{\max})$ becomes increasingly pronounced—aligning with the observation in Table 2. Overall, these results demonstrate that iteratively updating $\pi_{\text{ref}}$ effectively reduces reward disparity, thereby enabling smoother and more stable policy improvement.

**Ablation study on the choices of $\beta$:** Lastly, we conduct an ablation study to investigate the discrepancy between theoretical and empirical choices of the KL coefficient $\beta$. According to Theorem 3.5, a smaller $\beta$ is theoretically preferable, as regularization drives the policy away from optimality. To examine this, we consider three choices of $\beta$: $\{0.1, 0.03, 0.01\}$. Specifically, we adjust the exploration coefficient $\alpha$ in accordance with $\beta$ to keep $\alpha\beta$ constant, thereby ensuring that the scale of the exploration term's gradient remains stable. The results, summarized in Table 7 in Appendix K.5, reveal that $\beta = 0.03$ performs best, followed by $\beta = 0.1$, and lastly $\beta = 0.01$. This suggests that while smaller $\beta$ values are theoretically desirable, an excessively small $\beta$ can introduce instability during training, leading to suboptimal performance in practice.

# 5 Limitation & Conclusion

In this work, we propose SE-POPO, the first practical and provably sample-efficient online exploration algorithm for RLHF with a polynomial dependence on the reward scale. SE-POPO offers a strictly superior sample complexity guarantee in theory, while outperforming existing baselines in practice. One limitation of the approach is that SE-POPO does not extend to general preference models beyond Bradley–Terry model, particularly those where the preference is not necessarily monotonic. Future directions include investigating online exploration algorithms with minimal length exploitation [Singhal et al., 2023, Meng et al., 2024], extending our algorithms to token-level MDP [Xie et al., 2024, Zhong et al., 2024] and multi-turn RLHF settings [Shani et al., 2024, Gao et al., 2024, Xiong et al., 2024b].

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

# Contents

## A   Related Works

**RLHF and RLHF algorithms**   The current RLHF framework was first popularized by [Christiano et al., 2017], which served to direct the attention of the deep RL community to the preference-based feedback. Due to its significant success in LLM alignment [OpenAI, 2022, Touvron et al., 2023], RLHF has gained substantial interest and become one of the prominent research topics in recent years. The most widely adopted and standard RLHF framework, as described in [Ouyang et al., 2022, Touvron et al., 2023], consists of two primary stages: 1) optimizing a reward model using the preference dataset, and 2) refining the LLM policy using PPO [Schulman et al., 2017] based on the optimized reward model. While this RLHF framework has achieved tremendous success in the industry, its adoption by academic and open-source communities is challenging due to the essential limitations of PPO, such as issues with reproducibility [Choshen et al., 2019], hyperparameters sensitivity [Engstrom et al., 2020], and its significant computational resource requirements. Inspired by the limitations of this two-stage approach, a new line of research focuses on single-stage algorithms, including SLiC [Zhao et al., 2023], DPO [**?**], and its variants, such as IPO [Azar et al., 2024], SPPO [Wu et al., 2024], VPO [Cen et al., 2024], XPO [Xie et al., 2024], and SELM [Zhang et al., 2024]. These algorithms bypass the reward modeling step and learn a policy by optimizing a designed loss function on the preference dataset directly. It is observed that such algorithms are much more stable than PPO and achieve impressive performance on public benchmarks [Tunstall et al., 2023, Dubois et al., 2024, Zheng et al., 2023].

**Theoretical Study on RLHF**   The earliest theoretical frameworks for RLHF trace back to the dueling bandits literature [Yue et al., 2012, Saha and Gopalan, 2018, Bengs et al., 2021], along with studies considering tabular RL with finite state space [Xu et al., 2020, Novoseller et al., 2020] and linear RL or general function approximation RL with infinite state space [Pacchiano et al., 2021, Chen et al., 2022, Wu and Sun, 2023, Zhan et al., Das et al., 2024a, Wang et al., 2023]. Apart from the online setting, a substantial body of research focuses on offline RLHF [Zhu et al., 2023, Zhan et al., 2023, Ji et al., 2024, Liu et al., 2024], which leverages predetermined offline datasets with appropriate coverage conditions over the state-action space and can be considered complementary to our work. Although these studies offer sample complexity guarantees for RLHF, most algorithms are not scalable enough to be applicable to modern LLMs with large transformer architectures. For instance, Pacchiano et al. [2021], Das et al. [2024a] incorporate exploration bonuses tailored for linear models in the reward estimation. Chen et al. [2022], Zhan et al. [2023], Wang et al. [2023] rely on model-based function approximation and explicitly estimate the policy confidence set. These approaches fail to yield efficient or practical algorithms when applied to LLMs.

**Exploration for online LLM alignment**   Exploration in online RLHF has seen rapid development recently. Earlier attempts, such as online DPO [Guo et al., 2024] and iterative DPO [Xu et al., 2023, Dong et al., 2024, Xiong et al., 2024b], primarily rely on passive exploration, i.e. the inherent randomness of LLM policy, and lack explicit mechanisms to encourage diverse and exploratory responses. The importance of active exploration in RLHF has been highlighted by Dwaracherla et al. [2024]. Subsequently, Ye et al. [2024], Xiong et al. [2024a] propose algorithms with an active exploration mechanism and provide a sample complexity guarantees for online RLHF. However, these exploration strategies involve solving an intractable optimization problem, making them impractical to implement in LLM alignment. Notably, in these works, experiments are often conducted based on heuristic variants of the proposed algorithms, resulting in a significant gap between theory and practice. More recently, Cen et al. [2024], Xie et al. [2024], Zhang et al. [2024] introduce implementation-friendly and provably sample-efficient exploration algorithms for RLHF, which are most relevant to our work. All three papers are based on the common idea of augmenting the DPO loss with a *reward-based* optimistic bonus to encourage exploration. Among them, Zhang et al. [2024], Cen et al. [2024] mainly focus on the exploration under the contextual bandit formulation of RLHF, whereas Xie et al. [2024] provides analysis for the token-level MDP formulation. However, a significant limitation of these algorithms is that their sample complexity scales exponentially with $R_{\max}$, the scale of the reward function (see Asm. 2.1), which is highly inefficient in both theory and practice. Our algorithm becomes the first that remove such $\exp(R_{\max})$ dependency.

## B    Lightweight Implementation of SE-POPO

In this section, we demonstrate that the moving from (11) to (12) is virtually a "free-lunch" reduction, based on the following neat observation.

**Lemma B.1.** *Define* $H(r, \pi) = \underset{\substack{x \sim \rho \\ y' \sim \pi_{sam}(\cdot|x)}}{\mathbb{E}} \left[ \sigma \left( -\beta \log \frac{\pi(y'|x)}{\pi_{ref}(y'|x)} \right) \right]$ *, then for every* $r \in \mathcal{R}$*, we have*

$$|G(r, \pi(r)) - H(r, \pi(r))| \leq \frac{\beta}{2} \mathbb{E}_{x \sim \rho}[\mathbb{D}_{KL}(\pi_r^\star(\cdot|x)||\pi_{ref}(\cdot|x))],$$

*where* $\pi_r^\star = \arg\max_\pi \mathbb{E}_{x \sim \rho, y \sim \pi(\cdot|x)}[r(x, y)]$*.*

Lemma B.1 implies that the gap between using $G(r, \pi(r))$ and $H(r, \pi(r))$ scales with $\beta$ and the KL divergence between $\pi_r^\star$ and $\pi_{\text{ref}}$. In this case, given $\beta = o(1/\sqrt{T})$, replacing $G(r, \pi(r))$ with $H(r, \pi(r))$ in the optimization objective (10) still guarantees that Theorem 3.4 essentially holds, i.e.,

**Theorem B.2.** *By replacing* $G(r, \pi(r))$ *in the optimization objective (10) with* $H(r, \pi(r))$*, with probability* $1 - 2\delta$*,* POPO *guarantees that* $\text{Reg}_{pref}(\pi_{sam}, T) \leq$

$$\mathcal{O} \left( \sqrt{dR_{max}T \log \frac{TR_{max}}{d} \log \frac{|\mathcal{R}|}{\delta}} + d \exp(R_{max}) \log \frac{TR_{max}}{d} \log \frac{|\mathcal{R}|}{\delta} + \beta T C_{KL}' \right),$$

*where* $C_{KL}' = \max_{r \in \mathcal{R}} \mathbb{E}_{x \sim \rho} \left[ \mathbb{D}_{KL}(\pi_r^\star(\cdot|x)||\pi_{ref}(\cdot|x)) \right]$*.*

Theorem B.2 establishes a preference regret bound that is fundamentally consistent with Theorem 3.4, with the only difference being in the KL term. In particular, when $\beta$ is sufficiently small, Theorem B.2 reduces to Theorem 3.4 immediately. Therefore, assuming $C_{KL}'$ is well-bounded, it follows that the reward regret in Theorem 3.5 remains valid with the new exploration bonus.

## C    Supporting Lemmas

We now present several auxiliary lemmas that will be used in next section's proof.

**Lemma C.1.** *(MLE estimation error [Cen et al., 2024, Xie et al., 2024]) With probability at least* $1 - \delta$*, for all* $r \in \mathcal{R}$ *and* $t \in [T]$*, there is*

$$\ell(r^\star, \mathcal{D}_{t-1}) - \ell(r, \mathcal{D}_{t-1})$$
$$\leq -\frac{1}{2} \sum_{s=1}^{t-1} \mathbb{E}_{x \sim \rho, (y,y') \sim \pi_s \otimes \pi_{sam}(\cdot|x)} \left[ (\mathbb{P}^\star(y \succ y'|x) - \mathbb{P}_r(y \succ y'|x))^2 \right] + 2 \log \frac{|\mathcal{R}|}{\delta}.$$

**Lemma C.2.** *Define*

$$\mathcal{R}(\mathcal{D}) = \left\{ r \in \mathcal{R} \mid \ell(r, \mathcal{D}) - \min_{r' \in \mathcal{R}} \ell(r', \mathcal{D}) \leq 2 \log \frac{|\mathcal{R}|}{\delta} \right\}.$$

*Conditioning on Lemma C.1, for all* $t \in [T]$*, there is* $r^\star \in \mathcal{R}(\mathcal{D}_t)$*.*

**Lemma C.3.** *Conditioning on Lemma C.2, for all* $t \in [T]$*, there is*

$$r_{t+1} = \arg\max_{r \in \mathcal{R}(\mathcal{D}_t)} \left\{ -\ell(r, \mathcal{D}_t) + \alpha G(r, \pi(r)) \right\}$$

**Lemma C.4.** *With probability at least* $1 - \delta$*, for all* $r \in \mathcal{R}$ *and* $t \in [T]$*, there is*

$$\sum_{s=1}^{t-1} \left( \mathbb{P}^\star(y_s \succ y_s' \mid x_s) - \mathbb{P}_r(y_s \succ y_s' \mid x_s) \right)^2$$
$$\leq 2 \sum_{s=1}^{t-1} \mathbb{E}_{x \sim \rho, (y,y') \sim \pi_s \otimes \pi_{sam}(\cdot|x)} \left[ (\mathbb{P}^\star(y \succ y' \mid x) - \mathbb{P}_r(y \succ y' \mid x))^2 \right] + \log \frac{|\mathcal{R}|}{\delta}.$$

# D Proof of Theorem 3.4

By the definition of $G$, there is

$$\text{Reg}_{\text{pref}}(\pi_{\text{sam}}, T) \leq \sum_{t=1}^{T}[G(r^\star, \pi^\star) - G(r^\star, \pi_t)]$$

$$= \underbrace{\sum_{t=1}^{T}[G(r^\star, \pi_\beta^\star) - G(r_t, \pi_t)]}_{\text{TERM 1}} + \underbrace{\sum_{t=1}^{T}[G(r_t, \pi_t) - G(r^\star, \pi_t)]}_{\text{TERM 2}} + \underbrace{\sum_{t=1}^{T}[G(r^\star, \pi^\star) - G(r^\star, \pi_\beta^\star)]}_{\text{TERM 3}}$$

where $\pi_\beta^\star = \arg\max_{\pi \in \Pi} J(r^\star, \pi)$ and $r_t$ represents the reward corresponding to $\pi_t$, i.e., $\pi_t = \arg\max_{\pi \in \Pi} J(r_t, \pi)$.

**Bounding TERM 1**   Notice that in objective (9), $\pi_t$ is completely dependent on $r_t$. In this regard, the function $G$ can be considered as a function that depends only on the reward. By Lemma C.2 and C.3, we have

$$-\ell(r^\star, \mathcal{D}_{t-1}) + \alpha G(r^\star, \pi_\beta^\star) \leq -\ell(r_t, \mathcal{D}_{t-1}) + \alpha G(r_t, \pi_t),$$

thus

$$G(r^\star, \pi_\beta^\star) - G(r_t, \pi_t) \leq \frac{1}{\alpha}[\ell(r^\star, \mathcal{D}_{t-1}) - \ell(r_t, \mathcal{D}_{t-1})].$$

By Lemma C.1, it holds that

$$\text{TERM 1} \leq -\frac{1}{2\alpha}\sum_{t=1}^{T}\sum_{s=1}^{t-1}\mathbb{E}_{x\sim\rho,(y,y')\sim\pi_s\otimes\pi_{\text{sam}}(\cdot|x)}\left[(\mathbb{P}^\star(y \succ y'|x) - \mathbb{P}_{r_t}(y \succ y'|x))^2\right] + \frac{2}{\alpha}T\log\frac{|\mathcal{R}|}{\delta}.$$

**Bounding TERM 2**   By Assumption 3.3, we can rewrite TERM 2 into

TERM 2

$$= \sum_{t=1}^{T}\mathbb{E}_{x\sim\rho,(y,y')\sim\pi_t\otimes\pi_{\text{sam}}(\cdot|x)}[\mathbb{P}_{r_t}(y \succ y'|x) - \mathbb{P}^\star(y \succ y'|x)]$$

$$= \sum_{t=1}^{T}\mathbb{E}_{x\sim\rho,(y,y')\sim\pi_t\otimes\pi_{\text{sam}}(\cdot|x)}[\sigma(r_t(x,y) - r_t(x,y')) - \sigma(r^\star(x,y) - r^\star(x,y'))]$$

$$\leq \sum_{t=1}^{T}\mathbb{E}_{x\sim\rho,(y,y')\sim\pi_t\otimes\pi_{\text{sam}}(\cdot|x)}[\sigma(\langle\theta_t, \phi(x,y,y')\rangle) - \sigma(\langle\theta^\star, \phi(x,y,y')\rangle)]$$

$$\leq \sum_{t=1}^{T}\mathbb{E}_{x\sim\rho,(y,y')\sim\pi_t\otimes\pi_{\text{sam}}(\cdot|x)}\left[\min\left\{\max_{v\in[-|\theta^\star,\phi(x,y,y')|,|\theta_t,\phi(x,y,y')|]}\dot\sigma(v)|\langle\theta_t - \theta^\star, \phi(x,y,y')\rangle|, 1\right\}\right]$$

$$\leq \sum_{t=1}^{T}\mathbb{E}_{x\sim\rho,(y,y')\sim\pi_t\otimes\pi_{\text{sam}}(\cdot|x)}[\min\{\exp(|\langle\theta_t - \theta^\star, \phi(x,y,y')\rangle|)\dot\sigma(|\langle\theta^\star, \phi(x,y,y')\rangle|)|\langle\theta_t - \theta^\star, \phi(x,y,y')\rangle|, 1\}]$$

$$\leq \underbrace{\sum_{t=1}^{T}\mathbb{E}_{x\sim\rho,(y,y')\sim\pi_t\otimes\pi_{\text{sam}}(\cdot|x)}\left[\min\left\{|\langle\theta_t - \theta^\star, \phi(x,y,y')\rangle|^2, 1\right\}\right]}_{\text{TERM 2(1)}}$$

$$+ \underbrace{3\sum_{t=1}^{T}\mathbb{E}_{x\sim\rho,(y,y')\sim\pi_t\otimes\pi_{\text{sam}}(\cdot|x)}[\dot\sigma(|\langle\theta^\star, \phi(x,y,y')\rangle|)|\langle\theta_t - \theta^\star, \phi(x,y,y')\rangle|]}_{\text{TERM 2(2)}}.$$

Here, $\dot\sigma$ represents the derivative of the sigmoid function. The second inequality uses $\sigma(a) - \sigma(b) \leq \min\left(\max_{v\in[-|b|,|a|]}\dot\sigma(v)|a - b|, 1\right)$, which holds since $\dot\sigma$ is symmetric and $\sigma$ is bounded above by

1. The third inequality is because $\max_{v\in[-|b|,|a|]} \dot\sigma(v) \leq \dot\sigma(b)\exp(|a-b|)$, which follows from the exponential tail behavior of the logistic derivative. The last inequality is due to $\min\{\exp(a)b, 1\} \leq \min\{a^2, 1\} + \exp(1)b$ for any $a, b \geq 0$. Denote by

$$W_t = \theta_t - \theta^\star, \; X_t = \phi(x_t, y_t, y_t'), \; w_t = \dot\sigma(|\langle\theta^\star, \phi(x_t, y_t, y_t')\rangle|),$$

$$Y_t = w_t X_t, \; \Lambda_t = \epsilon\mathbf{I} + \sum_{s=1}^{t-1} X_s X_s^\top, \; \Sigma_t = \epsilon\mathbf{I} + \sum_{s=1}^{t-1} Y_s Y_s^\top,$$

for some $\epsilon > 0$. We first focus on bounding TERM 2(1). By the definition of $\theta_t$, it suffices to note that

$$\|W_t\|_{\Lambda_t}^2 = \epsilon\|W_t\|^2 + \sum_{s=1}^{t-1} \langle W_t, X_s\rangle^2$$

$$\leq \epsilon R_{\max}^2 + \sum_{s=1}^{t-1}((r_t(x_s, y_s) - r_t(x_s, y_s')) - (r^\star(x_s, y_s) - r^\star(x_s, y_s')))^2$$

$$\leq \epsilon R_{\max}^2 + (1 + \exp(R_{\max}))\sum_{s=1}^{t-1}\left(\mathbb{P}^\star(y_s \succ y_s'|x_s) - \mathbb{P}_{r_t}(y_s \succ y_s'|x_s)\right)^2.$$

By Lemma C.1, C.3 and C.4, there is

$$\sum_{s=1}^{t-1}\left(\mathbb{P}^\star(y_s \succ y_s'|x_s) - \mathbb{P}_{r_t}(y_s \succ y_s'|x_s)\right)^2$$

$$\leq 2\sum_{s=1}^{t-1}\mathbb{E}_{x\sim\rho,(y,y')\sim\pi_s\otimes\pi_{\mathrm{sam}}(\cdot|x)}\left[\left(\mathbb{P}^\star(y \succ y' \mid x) - \mathbb{P}_{r_t}(y \succ y' \mid x)\right)^2\right] + \log\frac{|\mathcal{R}|}{\delta}$$

$$\leq 9\log\frac{|\mathcal{R}|}{\delta} + 4\ell(r_t, \mathcal{D}_{t-1}) - 4\ell(r^\star, \mathcal{D}_{t-1})$$

$$\leq 9\log\frac{|\mathcal{R}|}{\delta} + 4\ell(r_t, \mathcal{D}_{t-1}) - 4\min_{r'\in\mathcal{R}}\ell(r', \mathcal{D}_{t-1}) \leq 17\log\frac{|\mathcal{R}|}{\delta}.$$

Thus we have

$$\|W_t\|_{\Lambda_t}^2 \leq \epsilon R_{\max}^2 + 17(1 + \exp(R_{\max}))\log\frac{|\mathcal{R}|}{\delta}, \; \forall t,$$

which means

TERM 2(1)

$$\leq \sum_{t=1}^T \mathbb{E}_{x\sim\rho,(y,y')\sim\pi_t\otimes\pi_{\mathrm{sam}}(\cdot|x)}\left[\min\left\{\|W_t\|_{\Lambda_t}^2\|X_t\|_{\Lambda_t^{-1}}^2, 1\right\}\right]$$

$$\leq \left(\epsilon R_{\max}^2 + 4\exp(R_{\max})\log\frac{|\mathcal{R}|T}{\delta}\right)\left(\sum_{t=1}^T \mathbb{E}_{x\sim\rho,(y,y')\sim\pi_t\otimes\pi_{\mathrm{sam}}(\cdot|x)}\left[\min\left\{\|X_t\|_{\Lambda_t^{-1}}^2, 1\right\}\right]\right).$$

To proceed, we recall the elliptical potential lemma.

**Lemma D.1.** *([Abbasi-Yadkori et al., 2011], Lemma 11) Let $\{X_t\}$ be a sequence in $\mathbb{R}^d$ and $\Lambda_0 \in \mathbb{R}^{d\times d}$ a positive definite matrix. Define $\Lambda_t = \Lambda_0 + \sum_{s=1}^{t-1} X_s X_s^\top$, if $\|X_t\|_2 \leq L$ for all $t$, there is*

$$\sum_{t=1}^T \min\left\{1, \|X_t\|_{\Lambda_t^{-1}}^2\right\} \leq 2(d\log(trace(\Lambda_0) + TL^2/d) - \log\det(\Lambda_0)).$$

Applying this lemma we can get

$$\sum_{t=1}^T \min\left\{1, \|X_t\|_{\Lambda_t^{-1}}^2\right\} \leq 2d\log\left(1 + \frac{4TR_{\max}^2/d}{\epsilon}\right) := d(\epsilon).$$

Thus
$$\sum_{t=1}^{T} \mathbb{E}_{x\sim\rho,(y,y')\sim\pi_t\otimes\pi_{\text{sam}}(\cdot|x)} \left[ \min\left\{ \|X_t\|_{\Lambda_t^{-1}}, 1 \right\} \right] = \mathbb{E}_{\{x_t,y_t,y_t'\}_{t=1}^T} \left[ \sum_{t=1}^{T} \min\left\{ \|X_t\|_{\Lambda_t^{-1}}^2, 1 \right\} \right] \le d(\epsilon)$$
and
$$\text{TERM } 2(1) \le \epsilon d(\epsilon) R_{\max}^2 + 17 d(\epsilon)(1 + \exp(R_{\max})) \log \frac{|\mathcal{R}|}{\delta}$$

Now we start to bound TERM 2(2). We decompose the term into

$$\text{TERM } 2(2) = \mathbb{E}_{\{x_t,y_t,y_t'\}_{t=1}^T} \left[ \sum_{t=1}^{T} |\langle W_t, Y_t \rangle| \right]$$

$$= \mathbb{E}_{\{x_t,y_t,y_t'\}_{t=1}^T} \left[ \sum_{t=1}^{T} |\langle W_t, Y_t \rangle| \mathbb{1}\{ \|Y_t\|_{\Sigma_t^{-1}} \le 1 \} \right]$$

$$+ \mathbb{E}_{\{x_t,y_t,y_t'\}_{t=1}^T} \left[ \sum_{t=1}^{T} |\langle W_t, Y_t \rangle| \mathbb{1}\{ \|Y_t\|_{\Sigma_t^{-1}} > 1 \} \right]. \qquad (13)$$

Now we control the term terms in (13) respectively.

- The first term is bounded by

$$\sum_{t=1}^{T} |\langle W_t, Y_t \rangle| \mathbb{1}\{ \|Y_t\|_{\Sigma_t^{-1}} \le 1 \}$$

$$\le \sum_{t=1}^{T} \|W_t\|_{\Sigma_t} \|Y_t\|_{\Sigma_t^{-1}} \mathbb{1}\{ \|Y_t\|_{\Sigma_t^{-1}} \le 1 \}$$

$$\le \sum_{t=1}^{T} \|W_t\|_{\Sigma_t} \min\left\{ 1, \|Y_t\|_{\Sigma_t^{-1}} \right\}$$

$$= \sum_{t=1}^{T} \left[ \epsilon\|W_t\|^2 + \sum_{s=1}^{t-1} \langle W_t, Y_s \rangle^2 \right]^{1/2} \left[ \min\left\{ 1, \|Y_t\|_{\Sigma_t^{-1}}^2 \right\} \right]^{1/2}$$

$$\le \left\{ \sum_{t=1}^{T} \left[ \epsilon\|W_t\|^2 + \sum_{s=1}^{t-1} \langle W_t, Y_s \rangle^2 \right] \right\}^{1/2} \left\{ \sum_{t=1}^{T} \min\left\{ 1, \|Y_t\|_{\Sigma_t^{-1}}^2 \right\} \right\}^{1/2}$$

$$\le \sqrt{d(\epsilon)\epsilon T R_{\max}^2} + \sqrt{d(\epsilon)} \left\{ \sum_{t=1}^{T} \sum_{s=1}^{t-1} \langle W_t, Y_s \rangle^2 \right\}^{1/2}$$

$$\le \sqrt{d(\epsilon)\epsilon T R_{\max}^2} + \frac{d(\epsilon)}{2\mu} + \frac{\mu}{2} \sum_{t=1}^{T} \sum_{s=1}^{t-1} \langle W_t, Y_s \rangle^2,$$

where the third inequality is due to Cauchy–Schwarz inequality, the fourth inequality is because $\sqrt{a+b} \le \sqrt{a} + \sqrt{b}$, and the last inequality is by Young's inequality.

- The second term is bounded by applying Lemma D.1, i.e.,

$$\sum_{t=1}^{T} |\langle W_t, Y_t \rangle| \mathbb{1}\{ \|Y_t\|_{\Sigma_t^{-1}} > 1 \} \le R_{\max} \sum_{t=1}^{T} \mathbb{1}\{ \|Y_t\|_{\Sigma_t^{-1}} > 1 \}$$

$$\le R_{\max} \sum_{t=1}^{T} \min\left\{ 1, \|Y_i\|_{\Sigma_i^{-1}} \right\} \le R_{\max} d(\epsilon).$$

Summing up the two terms we arrive at

$$\sum_{t=1}^{T} |\langle W_t, Y_t \rangle| \le R_{\max} d(\epsilon) + \sqrt{d(\epsilon)\epsilon T R_{\max}^2} + \frac{d(\epsilon)}{2\mu} + \frac{\mu}{2} \sum_{t=1}^{T} \sum_{s=1}^{t-1} \langle W_t, Y_s \rangle^2.$$

thus

$$\text{TERM 2(2)} \leq R_{\max}d(\epsilon) + \sqrt{d(\epsilon)\epsilon T R_{\max}^2} + \frac{d(\epsilon)}{2\mu} + \frac{\mu}{2}\mathbb{E}_{\{x_t,y_t,y_t'\}_{t=1}^T}\left[\sum_{t=1}^T \sum_{s=1}^{t-1} \langle W_t, Y_s \rangle^2\right]$$

As expectation of sum is sum of expectation, we have

$$\mathbb{E}_{\{x_t,y_t,y_t'\}_{t=1}^T}\left[\sum_{t=1}^T \sum_{s=1}^{t-1} \langle W_t, Y_s \rangle^2\right] = \sum_{t=1}^T \sum_{s=1}^{t-1} \mathbb{E}_{x_s \sim \rho,(y_s,y_s') \sim \pi_s \otimes \pi_{\text{sam}}(\cdot|x)}\left[\langle W_t, Y_s \rangle^2\right]$$

$$= \sum_{t=1}^T \sum_{s=1}^{t-1} \mathbb{E}_{x_s \sim \rho,(y_s,y_s') \sim \pi_s \otimes \pi_{\text{sam}}(\cdot|x)}\left[\dot\sigma(\langle\theta^\star, X_s\rangle)^2 (\langle\theta_t, X_s\rangle - \langle\theta^\star, X_s\rangle)^2\right].$$

To proceed, we introduce an auxiliary lemma.

**Lemma D.2.** *For any $a, b \in [-R_{max}/2, R_{max}/2]$, there is*

$$\dot\sigma(a)|a - b| \leq 3R_{max}|\sigma(a) - \sigma(b)|.$$

By Lemma D.2, we have

$$\sum_{t=1}^T \sum_{s=1}^{t-1} \mathbb{E}_{x_s \sim \rho,(y_s,y_s') \sim \pi_s \otimes \pi_{\text{sam}}(\cdot|x)}\left[\dot\sigma(\langle\theta^\star, X_s\rangle)^2 (\langle\theta_t, X_s\rangle - \langle\theta^\star, X_s\rangle)^2\right]$$

$$\leq 3R_{\max}\sum_{t=1}^T \sum_{s=1}^{t-1} \mathbb{E}_{x_s \sim \rho, y_s \sim \pi_s(\cdot|x), y_s' \sim \pi_{\text{sam}}(\cdot|x)}\left[(\sigma(\langle\theta_t, X_s\rangle) - \sigma(\langle\theta^\star, X_s\rangle))^2\right]$$

$$= 3R_{\max}\sum_{t=1}^T \sum_{s=1}^{t-1} \mathbb{E}_{x \sim \rho, y \sim \pi_s(\cdot|x), y' \sim \pi_{\text{sam}}(\cdot|x)}\left[(\mathbb{P}_{r_t}(y \succ y'|x) - \mathbb{P}^\star(y \succ y'|x))^2\right].$$

Combining the above, we finally get

$$\text{TERM 2} \leq R_{\max}d(\epsilon) + \sqrt{d(\epsilon)\epsilon T R_{\max}^2} + \frac{d(\epsilon)}{2\mu} + \epsilon d(\epsilon)R_{\max}^2 + 17d(\epsilon)(1 + \exp(R_{\max}))\log\frac{|\mathcal{R}|}{\delta}$$

$$+ \frac{3\mu R_{\max}}{2}\sum_{t=1}^T \sum_{s=1}^{t-1} \mathbb{E}_{x \sim \rho, y \sim \pi_s(\cdot|x), y' \sim \pi_{\text{sam}}(\cdot|x)}\left[(\mathbb{P}_{r_t}(y \succ y'|x) - \mathbb{P}^\star(y \succ y'|x))^2\right].$$

**Bounding TERM 3** By the choice of $\pi_t$ in (9), we have $J(r^\star, \pi^\star) \leq J(r^\star, \pi_\beta^\star)$. This implies that

$$\mathbb{E}_{x \sim \rho, (y^\star, y) \sim \pi^\star \otimes \pi_\beta^\star(\cdot|x)}\left[r^\star(x, y^\star) - r^\star(x, y)\right] \leq \mathbb{E}_{x \sim \rho, (y^\star, y) \sim \pi^\star \otimes \pi_\beta^\star(\cdot|x)}\left[\beta \log\frac{\pi^\star(y^\star|x)}{\pi_{\text{ref}}(y^\star|x)} - \beta \log\frac{\pi_\beta^\star(y|x)}{\pi_{\text{ref}}(y|x)}\right].$$

The key observation is that for any $y' \in \mathcal{Y}$, there is

$$r^\star(x, y^\star) - r^\star(x, y) \geq 4[\mathbb{P}^\star(y^\star \succ y'|x) - \mathbb{P}^\star(y \succ y'|x)].$$

This is because $y^\star$ is always the best response, which means that $r^\star(x, y^\star) \geq r^\star(x, y)$ for sure. Moreover, the gradient of sigmoid function is less than $1/4$, thereby the gap between the preferences is at most $1/4$th of the gap between rewards. Using the inequality, we have

$$\mathbb{E}_{x \sim \rho, (y^\star, y, y') \sim \pi^\star \otimes \pi_\beta^\star \otimes \pi_{\text{sam}}(\cdot|x)}\left[\mathbb{P}^\star(y^\star \succ y'|x) - \mathbb{P}^\star(y \succ y'|x)\right]$$

$$\leq \frac{1}{4}\mathbb{E}_{x \sim \rho, (y^\star, y) \sim \pi^\star \otimes \pi_\beta^\star(\cdot|x)}\left[\beta \log\frac{\pi^\star(y^\star|x)}{\pi_{\text{ref}}(y^\star|x)} - \beta \log\frac{\pi_\beta^\star(y|x)}{\pi_{\text{ref}}(y|x)}\right]$$

$$\leq \frac{1}{4}\mathbb{E}_{x \sim \rho, y^\star \sim \pi^\star(\cdot|x)}\left[\beta \log\frac{\pi^\star(y^\star|x)}{\pi_{\text{ref}}(y^\star|x)}\right]$$

$$= \frac{\beta}{4}\mathbb{E}_{x \sim \rho}\left[\mathbb{D}_{\text{KL}}(\pi^\star(\cdot|x)||\pi_{\text{ref}}(\cdot|x))\right],$$

Thus we have $\text{TERM 3} \leq \mathcal{O}(\beta T \mathbb{E}_{x \sim \rho}\left[\mathbb{D}_{\text{KL}}(\pi^\star(\cdot|x)||\pi_{\text{ref}}(\cdot|x))\right])$.

**Finishing up** Combining `Term 1` and `Term 2`, with probability $1 - 2\delta$ and $\epsilon = \frac{1}{T}$, there is

$$\sum_{t=1}^{T} [G(r^\star, \pi_\beta^\star) - G(r^\star, \pi_t)] \le \mathcal{O}\left(\frac{1}{\alpha} T \log\frac{|\mathcal{R}|}{\delta} + \frac{d(\epsilon)}{\mu} + d(\epsilon) \exp(R_{\max}) \log\frac{|\mathcal{R}|}{\delta}\right).$$

as long as $\frac{3R_{\max}\mu}{2} \le \frac{1}{2\alpha}$. Setting $\alpha = \sqrt{\frac{d \log\frac{T}{d}}{R_{\max} T \log\frac{|\mathcal{R}|}{\delta}}}, \mu = \frac{1}{3}\sqrt{\frac{T \log\frac{|\mathcal{R}|}{\delta}}{R_{\max} d \log\frac{T}{d}}}$, we finally bound $\text{Reg}_{\text{pref}}(\pi_{\text{sam}}, T)$ by

$$\mathcal{O}\left(\sqrt{dR_{\max}T \log\frac{TR_{\max}}{d} \log\frac{|\mathcal{R}|}{\delta}} + d \exp(R_{\max}) \log\frac{TR_{\max}}{d} \log\frac{|\mathcal{R}|}{\delta} + \beta T C_{\text{KL}}\right),$$

Therefore, for $T \ge \tilde{\mathcal{O}}(d \exp(2R_{\max}) \log\frac{|\mathcal{R}|}{\delta}/R_{\max})$, we have

$$\mathbb{E}_{\substack{x \sim \rho \\ (y^\star, y, y') \sim \pi^\star \otimes \bar{\pi} \otimes \pi_{\text{sam}}(\cdot|x)}} \left[\mathbb{P}^\star(y^\star \succ y'|x) - \mathbb{P}^\star(y \succ y'|x)\right] = \frac{1}{T}\text{Reg}_{\text{pref}}(\pi_{\text{sam}}, T)$$

$$\le \tilde{\mathcal{O}}\left(\sqrt{\frac{dR_{\max} \log\frac{|\mathcal{R}|}{\delta}}{T}} + \beta C_{\text{KL}}\right),$$

which completes the proof.

# E    Proof of Theorem 3.5

For every $k = 1, \ldots, K$, by Theorem 3.4, with probability $1 - \delta$, there is

$$\mathbb{E}_{x \sim \rho, (y^\star, y, y') \sim \pi^\star \otimes \pi_{\text{sam}}^{k+1} \otimes \pi_{\text{sam}}^k(\cdot|x)} \left[\mathbb{P}^\star(y^\star \succ y'|x) - P^\star(y \succ y'|x)\right]$$

$$= \frac{\text{Reg}_{\text{pref}}(\pi_{\text{sam}}^k, T)}{T} \le \tilde{\mathcal{O}}\left(\sqrt{\frac{dR_{\max} \log\frac{|\mathcal{R}|}{\delta}}{T}} + \beta C_{\text{KL}}\right) \tag{14}$$

By Lemma 3.1, we have

$$\mathbb{E}_{x \sim \rho, (y^\star, y, y') \sim \pi^\star \otimes \pi_{\text{sam}}^{k+1} \otimes \pi_{\text{sam}}^k(\cdot|x)} \left[\mathbb{1}\{r^\star(x, y) - r^\star(x, y') \le 1\} \left[r^\star(x, y^\star) - r^\star(x, y)\right]\right]$$

$$\le \tilde{\mathcal{O}}\left(\sqrt{\frac{dR_{\max}^3 \log\frac{|\mathcal{R}|}{\delta}}{T}} + \beta R_{\max} C_{\text{KL}}\right) =: \text{Gap}(T) \tag{15}$$

For notation simplicity, we denote $r^\star(x, y) - r^\star(x, y')$ by $\Delta(x, y, y')$. To proceed, we note that

$$\mathbb{1}\{\Delta(x, y, y') \le 1\} \ge \mathbb{1}\{\Delta(x, y^\star, y) > \max(R_{\max} - k, 1)\}\mathbb{1}\{\Delta(x, y^\star, y') \le \max(R_{\max} - k + 1, 1)\}.$$

This is because when $\Delta(x, y^\star, y) > \max(R_{\max} - k, 1)$ and $\Delta(x, y^\star, y') \le \max(R_{\max} - k + 1, 1)$, we have

$$\Delta(x, y, y') = \Delta(x, y^\star, y') - \Delta(x, y^\star, y)$$
$$\le \max(R_{\max} - k + 1, 1) - \max(R_{\max} - k, 1) \le 1.$$

In this regard, given $r^\star(x, y^\star) - r^\star(x, y) \geq 0$ for sure, we have

$$\mathbb{E}_{x\sim\rho,(y^\star,y,y')\sim\pi^\star\otimes\pi_{\mathrm{sam}}^{k+1}\otimes\pi_{\mathrm{sam}}^k(\cdot|x)}\left[\mathbb{1}\{r^\star(x,y)-r^\star(x,y')\leq 1\}\left[r^\star(x,y^\star)-r^\star(x,y)\right]\right]$$

$$= \mathbb{E}_{x\sim\rho,(y^\star,y,y')\sim\pi^\star\otimes\pi_{\mathrm{sam}}^{k+1}\otimes\pi_{\mathrm{sam}}^k(\cdot|x)}\left[\mathbb{1}\{\Delta(x,y,y')\leq 1\}\Delta(x,y^\star,y)\right]$$

$$\geq \mathbb{E}_{x\sim\rho,(y^\star,y,y')\sim\pi^\star\otimes\pi_{\mathrm{sam}}^{k+1}\otimes\pi_{\mathrm{sam}}^k(\cdot|x)}\Big[\mathbb{1}\{\Delta(x,y^\star,y)>\max(R_{\max}-k,1)\}$$
$$\mathbb{1}\{\Delta(x,y^\star,y')\leq\max(R_{\max}-k+1,1)\}\Delta(x,y^\star,y)\Big]$$

$$\geq \mathbb{E}_{x\sim\rho,(y^\star,y,y')\sim\pi^\star\otimes\pi_{\mathrm{sam}}^{k+1}\otimes\pi_{\mathrm{sam}}^k(\cdot|x)}\Big[\mathbb{1}\{\Delta(x,y^\star,y)>\max(R_{\max}-k,1)\}$$
$$\mathbb{1}\{\Delta(x,y^\star,y')\leq\max(R_{\max}-k+1,1)\}\Big]$$

$$\geq \mathbb{E}_{x\sim\rho,(y^\star,y')\sim\pi^\star\otimes\pi_{\mathrm{sam}}^k(\cdot|x)}\left[\mathbb{1}\{\Delta(x,y^\star,y')\leq\max(R_{\max}-k+1,1)\}\right]$$
$$- \mathbb{E}_{x\sim\rho,(y^\star,y)\sim\pi^\star\otimes\pi_{\mathrm{sam}}^{k+1}}\left[\mathbb{1}\{\Delta(x,y^\star,y)\leq\max(R_{\max}-k,1)\}\right].$$

The second inequality is because the inner term is non-zero only if $\Delta(x,y^\star,y)>\max(R_{\max}-k,1)\geq 1$. Combining this with (15), with probability $1-K\delta$, there is

$$\mathbb{E}_{x\sim\rho,(y^\star,y)\sim\pi^\star\otimes\pi_{\mathrm{sam}}^{K+1}(\cdot|x)}\left[\mathbb{1}\{\Delta(x,y^\star,y)\leq\max(R_{\max}-K,1)\}\right]$$

$$\geq \mathbb{E}_{x\sim\rho,(y^\star,y')\sim\pi^\star\otimes\pi_{\mathrm{sam}}^K(\cdot|x)}\left[\mathbb{1}\{\Delta(x,y^\star,y')\leq\max(R_{\max}-K+1,1)\}\right]-\mathrm{Gap}(T)$$

$$\geq \mathbb{E}_{x\sim\rho,(y^\star,y')\sim\pi^\star\otimes\pi_{\mathrm{sam}}^1(\cdot|x)}\left[\mathbb{1}\{\Delta(x,y^\star,y')\leq\max(R_{\max},1)\}\right]-K\mathrm{Gap}(T)$$

$$= 1 - \tilde{\mathcal{O}}\left(K\sqrt{\frac{dR_{\max}^3\log\frac{|\mathcal{R}|}{\delta}}{T}}+K\beta R_{\max}C_{\mathrm{KL}}\right).$$

Setting $K=\lceil R_{\max}\rceil-1$, we achieve that

$$\mathbb{E}_{x\sim\rho,(y^\star,y)\sim\pi^\star\otimes\pi_{\mathrm{sam}}^{\lceil R_{\max}\rceil}}\left[\mathbb{1}\{\Delta(x,y^\star,y)>1\}\right]$$

$$\leq \tilde{\mathcal{O}}\left(\sqrt{\frac{dR_{\max}^5\log\frac{|\mathcal{R}|}{\delta}}{T}}+\beta R_{\max}^2 C_{\mathrm{KL}}\right).$$

This result implies that

$$\mathbb{E}_{x\sim\rho,(y,y')\sim\bar{\pi}\otimes\pi_{\mathrm{sam}}^{\lceil R_{\max}\rceil}}\left[\mathbb{1}\{\Delta(x,y,y')>1\}\right]$$

$$\leq \tilde{\mathcal{O}}\left(\sqrt{\frac{dR_{\max}^5\log\frac{|\mathcal{R}|}{\delta}}{T}}+\beta R_{\max}^2 C_{\mathrm{KL}}\right).$$

for all $\bar{\pi}$. In this regard, it suffices to note that $\pi_{\mathrm{sam}}^{\lceil R_{\max}\rceil}$ is a "good enough" sampler: it can return a response $y'$ such that $\Delta(x,y,y')\leq 1$ with high probability. Denote by $\bar{\pi}=\mathrm{POPO}(\pi_{\mathrm{ref}},\pi_{\mathrm{sam}}^{\lceil R_{\max}\rceil},T)$,

with probability $1 - \delta$, there is

$$
\mathbb{E}_{x\sim\rho,(y^\star,y)\sim\pi^\star\otimes\bar{\pi}(\cdot|x)} \left[r^\star(x,y^\star) - r^\star(x,y)\right]
$$

$$
= \mathbb{E}_{x\sim\rho,(y^\star,y,y')\sim\pi^\star\otimes\bar{\pi}\otimes\pi_{\text{sam}}^{\lceil R_{\max}\rceil}(\cdot|x)} \left[\mathbb{1}\{\Delta(x,y,y') \leq 1\}[r^\star(x,y^\star) - r^\star(x,y)]\right]
$$

$$
+ \mathbb{E}_{x\sim\rho,(y^\star,y,y')\sim\pi^\star\otimes\bar{\pi}\otimes\pi_{\text{sam}}^{\lceil R_{\max}\rceil}(\cdot|x)} \left[\mathbb{1}\{\Delta(x,y,y') > 1\}[r^\star(x,y^\star) - r^\star(x,y)]\right]
$$

$$
\leq \mathbb{E}_{x\sim\rho,(y^\star,y,y')\sim\pi^\star\otimes\bar{\pi}\otimes\pi_{\text{sam}}^{\lceil R_{\max}\rceil}(\cdot|x)} \left[\mathbb{1}\{\Delta(x,y,y') \leq 1\}[r^\star(x,y^\star) - r^\star(x,y)]\right]
$$

$$
+ R_{\max}\mathbb{E}_{x\sim\rho,(y,y')\sim\bar{\pi}\otimes\pi_{\text{sam}}^{\lceil R_{\max}\rceil}(\cdot|x)} \left[\mathbb{1}\{\Delta(x,y,y') > 1\}\right]
$$

$$
\leq \tilde{\mathcal{O}}\left(\sqrt{\frac{dR_{\max}^7 \log\frac{|\mathcal{R}|}{\delta}}{T}} + \beta R_{\max}^3 C_{\text{KL}}\right).
$$

Setting $\beta \leq o\left(\frac{1}{\sqrt{T}}\right), T = N/\lceil R_{\max}\rceil$ and resizing $\delta = \delta/\lceil R_{\max}\rceil$ immediately complete the proof.

## F    Proof of Theorem B.2

In the proof of Theorem 3.4, the only place where we use the condition that $\pi_{t+1}$ is the optimal solution to objective (11) is in the proof of bounding TERM 1. Therefore, it suffices to focus on TERM 1 itself. As Lemma C.2 and C.3 still hold, we have

$$
-\ell(r^\star, \mathcal{D}_{t-1}) + \alpha H(r^\star, \pi_\beta^\star) \leq -\ell(r_t, \mathcal{D}_{t-1}) + \alpha H(r_t, \pi_t),
$$

Using Lemma B.1, it suffices to note

$$
-\ell(r^\star, \mathcal{D}_{t-1}) + \alpha G(r^\star, \pi_\beta^\star) - \frac{\alpha\beta}{2}\max_{r\in\mathcal{R}}\mathbb{E}_{x\sim\rho}\left[\mathbb{D}_{\text{KL}}(\pi_r^\star(\cdot|x)||\pi_{\text{ref}}(\cdot|x))\right] \leq -\ell(r^\star, \mathcal{D}_{t-1}) + \alpha H(r^\star, \pi_\beta^\star)
$$

$$
-\ell(r_t, \mathcal{D}_{t-1}) + \alpha H(r_t, \pi_t) \leq -\ell(r_t, \mathcal{D}_{t-1}) + \alpha G(r_t, \pi_t) + \frac{\alpha\beta}{2}\max_{r\in\mathcal{R}}\mathbb{E}_{x\sim\rho}\left[\mathbb{D}_{\text{KL}}(\pi_r^\star(\cdot|x)||\pi_{\text{ref}}(\cdot|x))\right],
$$

thus

$$
G(r^\star, \pi_\beta^\star) - G(r_t, \pi_t) \leq \frac{1}{\alpha}[\ell(r^\star, \mathcal{D}_{t-1}) - \ell(r_t, \mathcal{D}_{t-1})] + \beta\max_{r\in\mathcal{R}}\mathbb{E}_{x\sim\rho}\left[\mathbb{D}_{\text{KL}}(\pi_r^\star(\cdot|x)||\pi_{\text{ref}}(\cdot|x))\right].
$$

This completes the proof.

## G    Proof of Lemma 3.1

Assuming $r^\star(x,y) \leq r^\star(x,y') + 1$. In this case, we note that

$$
P^\star(y \succ y'|x) = \frac{\exp(r^\star(x,y) - r^\star(x,y'))}{1 + \exp(r^\star(x,y) - r^\star(x,y'))} \leq \frac{e}{1+e} \leq \frac{3}{4}.
$$

Given this, it suffices to focus on the case where $P^\star(y^\star \succ y'|x) \leq 4/5$, otherwise

$$
P^\star(y^\star \succ y'|x) - P^\star(y \succ y'|x) \geq \frac{4}{5} - \frac{3}{4} \geq \frac{r^\star(x,y^\star) - r^\star(x,y)}{20R_{max}}.
$$

Similarly, since $P^\star(y^\star \geq y'|x) \geq 1/2$, it suffices to focus on the case where $P^\star(y \succ y'|x) \geq 9/20$, otherwise

$$
P^\star(y^\star \succ y'|x) - P^\star(y \succ y'|x) \geq \frac{1}{2} - \frac{9}{20} \geq \frac{r^\star(x,y^\star) - r^\star(x,y)}{20R_{max}}.
$$

In this way, we obtain certain constraints on the preferences $P^\star(y^\star \succ y'|x)$ and $P^\star(y \succ y'|x)$. This further leads to constraints on the differences in rewards, i.e.,

$$
0 \leq r^\star(x,y^\star) - r^\star(x,y') \leq \frac{3}{2}, \quad -\frac{1}{2} \leq r^\star(x,y) - r^\star(x,y') \leq 1.
$$

Thus, it suffices to focus on the interval $[-\frac{1}{2}, \frac{3}{2}]$. It is easily to see that

$$
\begin{aligned}
&P^\star(y^\star \succ y'|x) - P^\star(y \succ y'|x) \\
&= \sigma(r^\star(x, y^\star) - r^\star(x, y')) - \sigma(r^\star(x, y) - r^\star(x, y')) \\
&\geq \min_{\Delta \in \left[-\frac{1}{2}, \frac{3}{2}\right]} \nabla\sigma(\Delta)[r^\star(x, y^\star) - r^\star(x, y') - (r^\star(x, y) - r^\star(x, y'))] \\
&= \frac{r^\star(x, y^\star) - r^\star(x, y)}{20}.
\end{aligned}
$$

Combining the above we have

$$
r^\star(x, y^\star) - r^\star(x, y) \leq 20 R_{\max}[P^\star(y^\star \succ y'|x) - P^\star(y \succ y'|x)]
$$

with $r^\star(x, y) - r^\star(x, y') \leq 1$. This completes the proof.

# H   Proof of Lemma B.1

Fix $r \in \mathcal{R}$. Recall $\pi(r)$ is the optimal solution of the KL-regularized reward objective and $\pi_r^\star = \arg\max_\pi \mathbb{E}_{x \sim \rho, y \sim \pi(\cdot|x)}[r(x, y)]$. By the analysis of bounding TERM 3 in the proof of Theorem 3.4, we first note that

$$
G(r, \pi_r^\star) - \frac{\beta}{4}\mathbb{E}_{x \sim \rho}[\mathbb{D}_{\mathrm{KL}}(\pi_r^\star(\cdot|x)||\pi_{\mathrm{ref}}(\cdot|x))] \leq G(r, \pi(r)) \leq G(r, \pi_r^\star).
$$

It suffices to focus on $G(r, \pi_r^\star)$. Then, we have

$$
\begin{aligned}
G(r, \pi_r^\star) &= \mathbb{E}_{x \sim \rho, y \sim \pi_r^\star(\cdot|x), y' \sim \pi_{\mathrm{sam}}(\cdot|x)}\left[\sigma(r(x, y) - r(x, y'))\right] \\
&= \mathbb{E}_{x \sim \rho, y \sim \pi_r^\star(\cdot|x), y' \sim \pi_{\mathrm{sam}}(\cdot|x)}\left[\sigma\left(\beta \log \frac{\pi_r(y|x)}{\pi_{\mathrm{ref}}(y|x)} - \beta \log \frac{\pi_r(y'|x)}{\pi_{\mathrm{ref}}(y'|x)}\right)\right]
\end{aligned}
$$

where $\pi_r = \pi(r)$. Since here $y$ represents the response with the highest reward under $r$, it suffices to note that $\pi_r(y|x) \geq \pi_{\mathrm{ref}}(y|x)$. In this case, $\beta \log \frac{\pi_r(y|x)}{\pi_{\mathrm{ref}}(y|x)}$ can be bounded by $[0, \beta \log \frac{1}{\pi_{\mathrm{ref}}(y|x)}]$. By the smoothness of sigmoid function, there is

$$
\begin{aligned}
&\mathbb{E}_{x \sim \rho, y' \sim \pi_{\mathrm{sam}}(\cdot|x)}\left[\sigma\left(-\beta \log \frac{\pi_r(y'|x)}{\pi_{\mathrm{ref}}(y'|x)}\right)\right] \\
&\leq \mathbb{E}_{x \sim \rho, y \sim \pi_r^\star(\cdot|x), y' \sim \pi_{\mathrm{sam}}(\cdot|x)}\left[\sigma\left(\beta \log \frac{\pi_r(y|x)}{\pi_{\mathrm{ref}}(y|x)} - \beta \log \frac{\pi_r(y'|x)}{\pi_{\mathrm{ref}}(y'|x)}\right)\right] \\
&\leq \mathbb{E}_{x \sim \rho, y \sim \pi_r^\star(\cdot|x), y' \sim \pi_{\mathrm{sam}}(\cdot|x)}\left[\sigma\left(-\beta \log \frac{\pi_r(y'|x)}{\pi_{\mathrm{ref}}(y'|x)}\right) + \frac{\beta}{4} \log \frac{\pi_r(y|x)}{\pi_{\mathrm{ref}}(y|x)}\right] \\
&\leq \mathbb{E}_{x \sim \rho, y' \sim \pi_{\mathrm{sam}}(\cdot|x)}\left[\sigma\left(-\beta \log \frac{\pi_r(y'|x)}{\pi_{\mathrm{ref}}(y'|x)}\right)\right] + \frac{\beta}{4}\mathbb{E}_{x \sim \rho}[\mathbb{D}_{\mathrm{KL}}(\pi_r^\star(\cdot|x)||\pi_{\mathrm{ref}}(\cdot|x))]
\end{aligned}
$$

The last inequality is due to $q = \arg\max_p \sum_y q(y) \log p(y)$. Combining the above we can conclude

$$
\left|G(r, \pi(r)) - \mathbb{E}_{x \sim \rho, y' \sim \pi_{\mathrm{sam}}(\cdot|x)}\left[\sigma\left(-\beta \log \frac{\pi_r(y'|x)}{\pi_{\mathrm{ref}}(y'|x)}\right)\right]\right| \leq \frac{\beta}{2}\mathbb{E}_{x \sim \rho}[\mathbb{D}_{\mathrm{KL}}(\pi_r^\star(\cdot|x)||\pi_{\mathrm{ref}}(\cdot|x))].
$$

This completes the proof.

# I   Proof of Auxiliary Lemmas

## I.1   Proof of Lemma C.1

The proof refers to the proof of Lemma 2 in [Cen et al., 2024]. To begin with, there is

$$
\ell(r^\star, \mathcal{D}_{t-1}) - \ell(r, \mathcal{D}_{t-1}) = -\sum_{s=1}^{t-1} \log \frac{\mathbb{P}_{r^\star}(y_s^+ \succ y_s^-|x_s)}{\mathbb{P}_r(y_s^+ \succ y_s^-|x_s)}.
$$

Define
$$X_r^s = \log \frac{\mathbb{P}_{r^\star}(y_s^+ \succ y_s^- | x_s)}{\mathbb{P}_r(y_s^+ \succ y_s^- | x_s)}.$$

Recall a martingale exponential inequality.

**Lemma I.1.** *([Zhang, 2023], Theorem 13.2) Let $\{X_t\}_{t=1}^\infty$ be a sequence of random variables adapted to filtration $\{\mathcal{F}_t\}_{t=1}^\infty$. It holds with probability $1 - \delta$ such that for any $t \geq 1$,*

$$-\sum_{s=1}^t X_s \leq \sum_{s=1}^t \log \mathbb{E}[\exp(-X_s)|\mathcal{F}_{s-1}] + \log \frac{1}{\delta}.$$

Notice that $\{X_r^t\}_{t=1}^\infty$ is a sequence of random variables adapted to filtration $\{\mathcal{F}_t\}_{t=1}^\infty$ with $\mathcal{F}_t$ given by the $\sigma$-algebra of $\{(x_s, y_s^+, y_s^-) : s \leq t\}$. Applying the above lemma and taking a union bound among all $r \in \mathcal{R}$, we have with probability $1 - \delta$, for every $r \in \mathcal{R}$ and $t$, there is

$$-\frac{1}{2} \sum_{s=1}^{t-1} X_r^s \leq \sum_{s=1}^{t-1} \log \mathbb{E}\left[\exp\left(-\frac{1}{2} X_r^s\right) \Big| \mathcal{F}_{s-1}\right] + \log \frac{|\mathcal{R}|}{\delta}$$

$$\leq \sum_{s=1}^{t-1} \left(\mathbb{E}\left[\exp\left(-\frac{1}{2} X_r^s\right) \Big| \mathcal{F}_{s-1}\right] - 1\right) + \log \frac{|\mathcal{R}|}{\delta},$$

where the last inequality is due to $\log(1 + x) \leq x$ for all $x \geq -1$. To proceed, note that

$$\mathbb{E}\left[\exp\left(-\frac{1}{2} X_r^s\right) \Big| \mathcal{F}_{s-1}\right] \leq \mathbb{E}\left[\sqrt{\frac{\mathbb{P}_r(y_s^+ \succ y_s^- | x_s)}{\mathbb{P}_{r^\star}(y_s^+ \succ y_s^- | x_s)}} \Big| \mathcal{F}_{s-1}\right]$$

$$= \mathbb{E}_{x \sim \rho, (y,y') \sim \pi_s \otimes \pi_{\mathrm{sam}}(\cdot|x), (+,-) \sim \mathbb{P}_{r^\star}(\cdot|x,y,y')}\left[\sqrt{\frac{\mathbb{P}_r(y^+ \succ y^- | x)}{\mathbb{P}_{r^\star}(y^+ \succ y^- | x)}}\right]$$

$$= \mathbb{E}_{x \sim \rho, (y,y') \sim \pi_s \otimes \pi_{\mathrm{sam}}(\cdot|x)}\left[\sum_{(+,-)} \sqrt{\mathbb{P}_r(y^+ \succ y^- | x) \mathbb{P}_{r^\star}(y^+ \succ y^- | x)}\right]$$

$$= 1 - \frac{1}{2} \mathbb{E}_{x \sim \rho, (y,y') \sim \pi_s \otimes \pi_{\mathrm{sam}}(\cdot|x)}\left[\sum_{(+,-)} \left(\sqrt{\mathbb{P}_r(y^+ \succ y^- | x)} - \sqrt{\mathbb{P}_{r^\star}(y^+ \succ y^- | x)}\right)^2\right]$$

$$\leq 1 - \frac{1}{8} \mathbb{E}_{x \sim \rho, (y,y') \sim \pi_s \otimes \pi_{\mathrm{sam}}(\cdot|x)}\left[\sum_{(+,-)} \left(\mathbb{P}_r(y^+ \succ y^- | x) - \mathbb{P}_{r^\star}(y^+ \succ y^- | x)\right)^2\right]$$

$$= 1 - \frac{1}{4} \mathbb{E}_{x \sim \rho, (y,y') \sim \pi_s \otimes \pi_{\mathrm{sam}}(\cdot|x)}\left[\left(\mathbb{P}_r(y \succ y' | x_s) - \mathbb{P}_{r^\star}(y \succ y' | x_s)\right)^2\right],$$

where the second inequality is due to $|\sqrt{x} - \sqrt{y}| \geq |x - y|/2$ for any $x, y \in [0,1]$. The last equality is because $|\mathbb{P}_r(y \succ y' | x_s) - \mathbb{P}_{r^\star}(y \succ y' | x_s)| = |\mathbb{P}_r(y' \succ y | x) - \mathbb{P}_{r^\star}(y' \succ y | x)|$. Combining the above, we finally have

$$\ell(r^\star, \mathcal{D}_{t-1}) - \ell(r, \mathcal{D}_{t-1}) \leq -\frac{1}{2} \sum_{s=1}^{t-1} \mathbb{E}_{x \sim \rho, (y,y') \sim \pi_s \otimes \pi_{\mathrm{sam}}(\cdot|x)}\left[\left(\mathbb{P}_r(y \succ y' | x) - \mathbb{P}_{r^\star}(y \succ y' | x)\right)^2\right] + 2 \log \frac{|\mathcal{R}|}{\delta},$$

which completes the proof.

## I.2 Proof of Lemma C.2

Conditioning on the event in Lemma C.1, we have

$$\ell(r^\star, \mathcal{D}_{t-1}) - \ell(r, \mathcal{D}_{t-1}) \leq 2 \log \frac{|\mathcal{R}|}{\delta}$$

for all $r \in \mathcal{R}$ and $t \in [T]$. This completes the proof.

### I.3 Proof of Lemma C.3

For any $r \in \mathcal{R}$ satisfying $\ell(r, \mathcal{D}_t) - \min_{r' \in \mathcal{R}} \ell(r', \mathcal{D}_t) > 2 \log \frac{|\mathcal{R}|}{\delta}$, there is

$$\ell(r, \mathcal{D}_t) - \ell(r^\star, \mathcal{D}_t) > 2 \log \frac{|\mathcal{R}|}{\delta} - 2 \log \frac{|\mathcal{R}|}{\delta} = 0$$

since $\ell(r^\star, \mathcal{D}_{t-1}) - \min_{r' \in \mathcal{R}} \ell(r', \mathcal{D}_t) \leq 2 \log \frac{|\mathcal{R}|}{\delta}$ by Lemma C.2. Therefore,

$$\begin{aligned}
-\ell(r, \mathcal{D}_t) + G(r, \pi(r)) I(r, \mathcal{D}_t) = -\ell(r, \mathcal{D}_t) &< -\ell(r^\star, \mathcal{D}_t) \\
&< -\ell(r^\star, \mathcal{D}_t) + G(r^\star, \pi(r^\star)) \\
&\leq \max_{r' \in \mathcal{R}} \left\{ -\ell(r', \mathcal{D}_t) + G(r', \pi(r')) I(r', \mathcal{D}_t) \right\}.
\end{aligned}$$

The last inequality is due to $I(r^\star, \mathcal{D}_t) = 1$ by Lemma C.2. This implies that such $r \notin \mathcal{R}(\mathcal{D}_t)$ cannot be the optimal solution of (10). In this case, it suffices to focus on $r \in \mathcal{R}(\mathcal{D}_t)$. Consider $I(r^\star, \mathcal{D}_t) = 1$ for every $r \in \mathcal{R}(\mathcal{D}_t)$, we complete the proof.

### I.4 Proof of Lemma C.4

Let $\mathcal{F}_t$ be a filtration. Denote by $X_t = \left( \mathbb{P}^\star(y_t \succ y_t' \mid x_t) - \mathbb{P}_r(y_t \succ y_t' \mid x_t) \right)^2$ and $P_t = \mathbb{E}_{x \sim \rho, (y, y') \sim \pi_t \otimes \pi_{\text{sam}}(\cdot \mid x)} \left[ \left( \mathbb{P}^\star(y \succ y' \mid x) - \mathbb{P}_r(y \succ y' \mid x) \right)^2 \right]$, it suffices to note that $(X_t)_{t \in \mathbb{N}^+}$ is a sequence of non-negative random variables satisfying $\mathbb{E}[X_t \mid \mathcal{F}_{t-1}] = P_t$. We first have

$$\begin{aligned}
\mathbb{E}[\exp(X_t - 2P_t) \mid \mathcal{F}_{t-1}] &\leq \mathbb{E}[1 + (X_t - 2P_t) + (X_t - 2P_t)^2 \mid \mathcal{F}_{t-1}] \\
&= \mathbb{E}[1 - P_t + X_t^2 \mid \mathcal{F}_{t-1}] \leq 1,
\end{aligned}$$

where the first inequality is because $\exp(a) \leq 1 + a + a^2$ for $a \in [-1, 1]$ and the last is due to $X_t^2 \leq X_t$. Denote by $Y_t = \exp(\sum_{s=1}^t (X_s - 2P_s))$, it suffices to note that $Y_1, \ldots, Y_T$ is a non-negative supermartingale. By Ville's inequality, we immediately have

$$\mathbb{P} \left( \exists t, Y_t > \frac{1}{\delta} \right) \leq \delta,$$

which implies

$$\mathbb{P} \left( \exists t, \sum_{s=1}^t X_s > 2 \sum_{s=1}^t P_s + \log \frac{1}{\delta} \right) \leq \delta.$$

Taking a union bound on $r \in \mathcal{R}$ completes the proof.

### I.5 Proof of Lemma D.2

*Proof.* Without loss of generality, we assume $a \geq 0$. We prove by case analysis.

1. $(b \in [a - 1, a + 1])$:

$$|\sigma(a) - \sigma(b)| \geq \dot{\sigma}(a + 1)|a - b| \geq \frac{1}{3} \dot{\sigma}(a)|a - b|.$$

2. $(b \notin [a - 1, a + 1])$:

$$\begin{aligned}
|\sigma(a) - \sigma(b)| &= \left| \frac{1}{1 + \exp(a)} - \frac{1}{1 + \exp(b)} \right| \\
&\geq \left| \frac{1}{1 + \exp(a)} - \frac{1}{1 + \exp(a + 1)} \right| \\
&= \frac{1}{1 + \exp(a)} \frac{\exp(a + 1) - \exp(a)}{1 + \exp(a + 1)} \\
&\geq \frac{1}{3} \frac{1}{1 + \exp(a)} \frac{\exp(a)}{1 + \exp(a)} \\
&\geq \frac{1}{3} \dot{\sigma}(a) \geq \frac{1}{3} \dot{\sigma}(a) \frac{|b - a|}{R_{\max}}.
\end{aligned}$$

$\square$

## J Generalization beyond linear preference oracle

In this section, we extend Theorem 3.4 from the linear reward oracle to a more general preference oracle. To do this, we introduce a general complexity measure—preference-based generalized eluder coefficient (PGEC)—which aligns with the complexity measures definitions in prior works [Xie et al., 2024, Zhang et al., 2024].

**Definition J.1.** (Preference-based GEC) Given a reward class $\mathcal{R}$, we define the preference-based Generalized Eluder Coefficient (PGEC) as the smallest $d_{\text{PGEC}}$ such that there exist $B \in O(1)$, s.t. for any $T$, $\gamma > 0$, sequence of policies $\pi_t \in \Pi$ and rewards $r_t \in \mathcal{R}$ satisfying $\sum_{s=1}^{t-1} \left( \mathbb{P}^\star(y_s \succ y'_s|x_s) - \mathbb{P}_{r_t}(y_s \succ y'_s|x_s) \right)^2 \le \gamma$, we have

$$\sum_{t=1}^{T} \mathbb{E}_{x \sim \rho, (y,y') \sim \pi_s \otimes \pi_{\text{sam}}(\cdot|x)} \left[ \mathbb{P}_{r_t}(y \succ y'|x) - \mathbb{P}^\star(y \succ y'|x) \right]$$

$$\le \sqrt{d_{\text{PGEC}} \sum_{t=1}^{T} \sum_{s=1}^{t-1} \mathbb{E}_{x \sim \rho, (y,y') \sim \pi_s \otimes \pi_{\text{sam}}(\cdot|x)} \left[ \left( \mathbb{P}_{r_t}(y \succ y'|x) - \mathbb{P}^\star(y \succ y'|x) \right)^2 \right]} + \sqrt{d_{\text{PGEC}} T} + B\gamma$$

The definition of PGEC is an variant of the Generalized Eluder Coefficient (GEC) proposed in Definition 3.4 of Zhong et al. [2022]. Specifically, initializing $B = \tilde{\mathcal{O}}(d \exp(R_{\max}))$, it suffices to note $d_{\text{PGEC}} = d R_{\max}$ for the linear reward case, as our selected $r_t$ satisfies $\sum_{s=1}^{t-1} \left( \mathbb{P}^\star(y_s \succ y'_s|x_s) - \mathbb{P}_{r_t}(y_s \succ y'_s|x_s) \right)^2 \le \mathcal{O}(\log(|\mathcal{R}|/\delta))$ for every $t$. By leveraging Definition J.1, we can extend the proof of Theorem 3.4 beyond the linear reward oracle. The only required modification is in the proof for bounding TERM 2. Notice that

$$\text{TERM 2} = \sum_{t=1}^{T} \mathbb{E}_{x \sim \rho, y \sim \pi_t(\cdot|x), y' \sim \pi_{\text{sam}}(\cdot|x)} \left[ \mathbb{P}_{r_t}(y \succ y'|x) - \mathbb{P}^\star(y \succ y'|x) \right]$$

$$\le \sqrt{d_{\text{PGEC}} \sum_{t=1}^{T} \sum_{s=1}^{t-1} \mathbb{E}_{x \sim \rho, y \sim \pi_s(\cdot|x), y' \sim \pi_{\text{sam}}(\cdot|x)} \left[ \left( \mathbb{P}_{r_t}(y \succ y'|x) - \mathbb{P}^\star(y \succ y'|x) \right)^2 \right]} + \sqrt{d_{\text{PGEC}} T} + B\gamma$$

$$\le \frac{d_{\text{PGEC}}}{2\mu} + \frac{\mu}{2} \sum_{t=1}^{T} \sum_{s=1}^{t-1} \mathbb{E}_{x \sim \rho, y \sim \pi_s(\cdot|x), y' \sim \pi_{\text{sam}}(\cdot|x)} \left[ \left( \mathbb{P}_{r_t}(y \succ y'|x) - \mathbb{P}^\star(y \succ y'|x) \right)^2 \right] + \sqrt{d_{\text{PGEC}} T} + B\gamma,$$

which matches the bound of TERM 2 in Theorem 3.4. Hence, with Definition J.1, it suffices to say that POPO guarantees

$$\text{Reg}_{\text{pref}}(\pi_{\text{sam}}, T) \le \tilde{\mathcal{O}} \left( \sqrt{d_{\text{PGEC}} T \log \frac{|\mathcal{R}|}{\delta}} + \beta T C_{\text{KL}} \right),$$

which also implies that the sample complexity of SE-POPO can be bounded by $\tilde{\mathcal{O}} \left( \frac{d_{\text{PGEC}} R_{\max}^7 \log \frac{|\mathcal{R}|}{\delta}}{\epsilon^2} \right)$.

## K Experiments Details

### K.1 Implementation Details

The experiments were conducted on 4 x Nvidia A100 80G GPUs. The pseudocode of our algorithm's implementation is illustrated in Algorithm 3. In the implementation, we set $\pi_{\text{sam}} = \pi_t$ and use the chosen responses to simulate the on-policy responses. To accelerate training, following Dong et al. [2024], we do not restart from the initial model at each iteration but use the last-iteration model as the initial checkpoint. Moreover, following Zhang et al. [2024], we update $\pi_{\text{ref}} = \pi_{t+1}$ for each iteration to avoid performance regression. For the implementations of DPO and XPO, they differ from Algorithm 3 only in the optimization objectives: DPO does not include the exploration bonus (i.e., $\alpha = 0$), while XPO replaces the exploration bonus to $-\alpha \sum_{(x,y^1) \in \mathcal{D}_t} \log \frac{\pi(y^1|x)}{\pi_{\text{ref}}(y^1|x)}$.

**Algorithm 3** Practical Implementation of `SE-POPO`

---

**Input:** Reference policy $\pi_{\text{ref}}$, Prompt dataset $\mathcal{D}$, Iterations $T$
**for** $t = 1, \ldots, T$ **do**
    Set $\mathcal{D}_t$ as the $t$-th portion of $\mathcal{D}$ and generate $(y^1, y^2) \sim \pi_{\text{ref}}(\cdot|x)$ for each prompt $x \in \mathcal{D}_t$.
    Annotate responses $(x, y^1, y^2) \rightarrow (x, y^w, y^l)$.
    Optimize

$$\pi_{t+1} = \arg\max_{\pi} \sum_{(x, y_w, y_l) \in \mathcal{D}_t} \log \sigma \left( \beta \log \frac{\pi(y^w|x)}{\pi_{\text{ref}}(y^w|x)} - \beta \log \frac{\pi(y^l|x)}{\pi_{\text{ref}}(y^l|x)} \right)$$
$$+ \alpha \sum_{(x, y^2) \in \mathcal{D}_t} \sigma \left( -\beta \log \frac{\pi(y^2|x)}{\pi_{\text{ref}}(y^2|x)} \right)$$

    Update $\pi_{\text{ref}} \leftarrow \pi_{t+1}$.
**end for**

---

Across all three experiments, we use `Llama-3-8B-SFT` [4] as the base model, `RLHFlow-ultrafeedback` [5] dataset as the training prompt sets, and `GRM-Llama3-8B-rewardmodel-ft` [6] as the training preference model. For hyperparameters, we mainly follow the settings in Xie et al. [2024] and Zhang et al. [2024]. We set $\beta = 0.1$, use a global batch size of 128, use a learning rate of $5 \times 10^{-7}$ with cosine scheduling. For exploration coefficient $\alpha$, we employ a decreasing strategy across iterations as in Xie et al. [2024] and do a grid search for $\alpha$ in the first iteration over $\{0.1, 0.01, 0.001, 0.0001, 0.00001\}$. Based on the empirical performance on AlphcaEval benchmark, we finally select $\{1 \times 10^{-3}, 5 \times 10^{-4}, 0\}$ for `XPO` and $\{1 \times 10^{-1}, 5 \times 10^{-2}, 0\}$ for `SE-POPO` respectively.

### K.2 Experiment results with base model `Qwen2.5-7B-Instruct`

To demonstrate that our algorithm generalizes across different base models, we further conduct experiments using base model `Qwen2.5-7B-Instruct`. In this experiment, we directly use all the hyperparameters from the setup in Appendix K.1. The results are shown below.

| Model | IID Data | | Alpaca Data | |
|---|---|---|---|---|
| | WR | AvgR | WR | AvgR |
| Qwen2.5-7B-Instruct | - | -2.86 | - | -3.39 |
| DPO-iter1 | 61.7 | -1.93 | 64.2 | -2.20 |
| DPO-iter2 | 67.1 | -1.26 | 78.3 | -0.81 |
| DPO-iter3 | 71.7 | -0.55 | 84.4 | 0.66 |
| XPO-iter1 | 61.8 | -1.92 | 69.5 | -1.80 |
| XPO-iter2 | 68.5 | -1.14 | 81.2 | -0.27 |
| XPO-iter3 | 73.5 | **-0.19** | 86.0 | 0.97 |
| SE-POPO-iter1 | 62.5 | -1.86 | 68.4 | -1.72 |
| SE-POPO-iter2 | 69.2 | -1.09 | 81.7 | -0.10 |
| SE-POPO-iter3 | **74.1** | -0.21 | **86.6** | **1.18** |

Table 4: Performance with base mode `Qwen2.5-7B-Instruct`

As shown, `SE-POPO` consistently outperforms both `DPO` and `XPO` even after to switching to a different base model. This further validates the robustness and effectiveness of our algorithm.

---

[4] https://huggingface.co/RLHFlow/LLaMA3-SFT
[5] https://huggingface.co/datasets/RLHFlow/ultrafeedback_iter1, https://huggingface.co/datasets/RLHFlow/ultrafeedback_iter2, https://huggingface.co/datasets/RLHFlow/ultrafeedback_iter3
[6] https://huggingface.co/Ray2333/GRM-Llama3-8B-rewardmodel-ft

### K.3 Academic Benchmarks

For academic benchmarks, following Xie et al. [2024], we select tasks MMLU Hendrycks et al. [2020], AGIEval Zhong et al. [2023], ANLI Nie et al. [2019], GPQA Rein et al. [2023], GSM8K Cobbe et al. [2021], WinoGrande Sakaguchi et al. [2019], TruthfulQA Lin et al. [2022], ARC Challenge Clark et al. [2018] and HellaSwag Zellers et al. [2019] as the benchmarks. The results are proposed in Table 5. It can be observed that with increasing iterations, both SE-POPO and other baselines may degrade on certain benchmarks, which is known as the alignment tax Askell et al. [2021], **?**], Lin et al. [2024]. Nevertheless, the evaluation result suggests that our method exhibits no additional degradation compared to DPO and XPO, while still effectively improving the base model across most benchmarks.

Table 5: Performance comparison across academic benchmarks

| Model | MMLU | AGIE | ANLI | GPQA | GSM8K | WINOG | TRUTH | ARC | HELLA |
|---|---|---|---|---|---|---|---|---|---|
| Llama-3-8B-SFT | 62.56 | 39.36 | 41.80 | 32.37 | 71.80 | 75.93 | 53.46 | 56.14 | 59.91 |
| DPO-iter1 | 62.75 | 40.32 | 44.00 | **32.81** | 76.64 | 76.24 | 56.18 | 55.97 | 79.58 |
| DPO-iter2 | 63.01 | 41.00 | 44.90 | 30.80 | 77.86 | 76.40 | 57.59 | 55.63 | 80.05 |
| DPO-iter3 | 63.11 | 41.56 | **46.90** | 31.25 | 77.55 | 76.16 | **59.48** | 54.78 | 80.33 |
| XPO-iter1 | 62.65 | 40.38 | 43.90 | 32.37 | 76.35 | 76.56 | 56.17 | 55.97 | 79.64 |
| XPO-iter2 | 63.14 | 41.38 | 45.70 | 31.25 | 77.33 | 76.95 | 58.58 | 55.38 | 80.29 |
| XPO-iter3 | 63.09 | 41.65 | 46.10 | 31.03 | **78.24** | **77.19** | 59.43 | 54.95 | 80.43 |
| POPO-iter1 | 62.80 | 40.45 | 44.00 | 32.37 | 76.80 | 76.00 | 56.21 | **56.14** | 79.80 |
| POPO-iter2 | 62.86 | 41.39 | 45.10 | 31.70 | 77.48 | 76.87 | 57.75 | 54.95 | 80.27 |
| POPO-iter3 | **63.13** | **41.68** | 45.60 | 31.92 | 77.63 | 76.63 | 59.14 | 54.35 | **80.67** |

### K.4 XPO theoretical implementation

| Model | IID Data | | Alpaca Data | | AE2 LC | MT-Bench | Avg. Len. (in AE2) |
|---|---|---|---|---|---|---|---|
| | WR | AvgR | WR | AvgR | | | |
| XPO-theory-iter1 | 62.6 | -4.40 | 78.3 | -5.79 | - | - | 1674 |
| XPO-theory-iter2 | 68.8 | -3.37 | 87.5 | -3.79 | - | - | 1886 |
| XPO-theory-iter3 | 71.7 | -2.62 | 91.0 | -1.21 | 30.70 | 7.91 | 2183 |

Table 6: Performance of XPO theoretical implementation

### K.5 Choices of KL-regularized coefficient $\beta$

| Model | IID Data | | Alpaca Data | |
|---|---|---|---|---|
| | WR | AvgR | WR | AvgR |
| SE-POPO-Beta-1e-1-iter1 | 62.5 | -4.32 | 80.0 | -5.68 |
| SE-POPO-Beta-1e-1-iter2 | 68.2 | -3.15 | 89.1 | -2.45 |
| SE-POPO-Beta-1e-1-iter3 | 73.3 | -2.03 | 92.4 | 0.61 |
| SE-POPO-Beta-3e-2-iter1 | 62.3 | -4.27 | 80.6 | -5.51 |
| SE-POPO-Beta-3e-2-iter2 | 70.0 | -3.10 | 88.6 | -2.49 |
| SE-POPO-Beta-3e-2-iter3 | 72.9 | -2.01 | 93.2 | 0.83 |
| SE-POPO-Beta-1e-2-iter1 | 62.8 | -4.32 | 78.7 | -5.70 |
| SE-POPO-Beta-1e-2-iter2 | 67.5 | -3.23 | 89.4 | -2.65 |
| SE-POPO-Beta-1e-2-iter3 | 72.0 | -2.10 | 92.3 | 0.54 |

Table 7: Performance across $\beta = \{0.1, 0.03, 0.01\}$

