# OpenReview forum: "Avoiding exp(R) scaling in RLHF through Preference-based Exploration"
_NeurIPS.cc/2025/Conference — NeurIPS 2025 poster_

### Official Review · Reviewer_rqsZ · 2025-06-28

**Clarity:** 3
**Significance:** 2
**Originality:** 3
**Rating:** 4
**Confidence:** 4

**Summary:**

This work proposes a solution to a known problem in RL in the context of modern LLM post-training: the exponential scaling of reward uncertainty as a function of the max possible reward difference between a preferred and dispreferred responses (eg. $\mathcal{O}(exp(R_{max}))$). Their algorithm operates by iteratively improving the policy used to sample the dispreferred responses, thereby closing the reward gap, and they argue that one can do so in $\mathcal{O}(R_{max})$ iterations of the algorithm's subroutine for improving the sampling policy. They theoretically derive how their algorithm implies their improved bound. Finally, they conduct standard preference tuning experiments to show that in practice, their method outperforms two existing baselines over three rounds of policy optimization over the same prompt dataset.

**Questions:**

### Questions
1. What is the reward-based exploration bonus $J(r, \pi)$? This is presented without definition and is important for later contrasting with the preference based regret defined in terms of $G(r, \pi)$ and $I(r, D)$. (If necessary, just present one concrete expression from a single prior work that the authors prefer.)
2. Is the preference based regret posed in Eq. 8 a novel definition in this work? Regardless, please style it as  definition and if drawn from literature, cite it.
3. Can the authors measure and report the empirical worst case reward gaps between $y$ and $y'$ for the datasets under all three methods?

Why is Q3 important for the paper? If the worst case gap between the preferred and dispreferred sample rewards under the same reward model is in fact reduced in practice this is an interesting observation to help connect the proposed theoretical improvement to the empirical improvement. It will not prove causality, but currently there is no evidence presented that the theory is related to the experiments. the algorithm could just be better for some other correlated reason.


### Minor/Writing Suggestions
- DPO is not just a way to avoid the explicit reward model, but it was also first posed in the offline setting where y and y' will be from a reference set. The notation around Eq 5 should note that canonically, y and y' come from D_train but that online variations of DPO allow one to continue to sample y,y' \sim theta(\cdot|x). Maybe the authors intend to immediately identify a fixed dataset of positive completions with generations from an optimal model? and negative completions with responses from a weak model (which need not necessarily be the current model though)? In any case, this distinction about offline setting of canonical DPO ([Rafailov) should be clarified.
- L131,L159 "optimism" is noun, so instead use the adjective "optimistic RLHF" to describe the concept of RLHF algorithms that factor in the principle of optimism.
- Please format Lemma 3.1 on its own line as it's difficult to parse as written. Also, the proof is central to the contribution of the work; the proposed algorithm relies on the surrogate minimization actually improving the exponential scaling problem posed. A sketch, poentially only in prose if necessary, should be presented in the main body.

**Ethical Concerns:**

["NO or VERY MINOR ethics concerns only"]

**Final Justification:**

See response.

**Limitations:**

- It is not clear that the empirical results support the theory, and without this demonstrated mechanistic connection, it is difficult to predict the generalizability of the method. Whereas, if the connection were clear, then any scenario where the exponential bound was relevant would be expected to also benefit from the application of the new algorithm.
- Generally, experiments on at least one other model family for the main policy, one other dataset, and one other reward model would help prove the robustness of the results, but I understand this is primarily a theory paper.

**Quality:**

3

**Strengths And Weaknesses:**

### Strengths

1. Presentation of the preliminaries are clear (but they use too much space given the lack of details in the actual contributions sections that follow.)

2. Presentation of the key problem statement is clear, but I implicitly read the bolded L141 as "How should we sample the responses y and y' in online RLHF? ..._to minimize reward uncertainty?_" as the section sets this up quite well.

(Authors don't need to imply that this is the only and central unanswered question in online RLHF as that would be an overly broad scope. Rather, assuming this posed bound is indeed an issue that significantly hampers overall learning performance, they have developed a theoretically grounded solution.)


### Weaknesses
1. Core elements of the theoretical setup and the claims made are not explained in clear enough detail in the main body, which is a problem, since this is a theory paper.
    1. An explanation for how Eq. 9 can be interpreted as encouraging exploration is missing as well as even a statement or sketch of how optimizing Eq. 9 minimizes Eq. 8. Forgive if this is obvious, but again as a theory paper, the core material seems to be lacking some key details that the reader needs to understand in a particular order to buy the contribution during a read-through.
    2.  Immediately dropping the actual bonus term from Eq.11 to Eq.12 seems premature. There exist frameworks to do this type of training efficiently (or even inefficiently). It should at least be shown that this is not a critical change. The concern stems how only increasing the second term of the original expression (just the second line of Eq 12) is somehow equivalent to incentivizing exploration in the same way as in the original objective. Maxing the neg log of the difference between the current policy and the reference policy (making them closer) on a _dispreferred_ sample has unclear utility. It is in fact the missing term where we max the difference between the current policy and ref policy on newly sampled positive completions that constitutes useful exploration of pushing the current policy away from the reference in the direction of potentially better responses. Imagine the pathological case of ending up with a policy such that for negative samples y' both the current policy and the reference policy assign equivalent likelihoods. Eg. solutions to the proposed optimization include identical policies or two non-identical, but otherwise uninteresting polices that both assign similar likelihoods to a given low reward response. Note that the proof in Appendix B is intended to show that Thm 3.5 still holds approximately when substituting with Eq. 12. However, the concern raised here is a separate point regardless of whether the bound holds or not and connected to further questions about what the alternate training algorithm is actually doing in practice (see below).

2. The empirical results are not particularly strong, but more importantly do not _directly_ support the theoretical claims of the paper; there is no demonstrated link between the overall performance improvement on the benchmarks and the key focus of the theory which centers to reward gaps and $R_{max}$.
    1. While the chosen experimental setup is standard practice (Table 1) the improvement of SE-POPO over other methods is relatively small.
    2. Since the argument is for sample complexity improvement (not necessarily absolute improvement at any specific budget) what we expect to see is evidence that as a function of available samples the rate of improvement for the proposed method is better. To be sure that the lead shown is reliable, it is better if experiments are taken to convergence, or peak WR and AvgR is achieved for the method given the starting policy and dataset. Does SE-POPO achieve 92.4 or 40.12 WR faster than DPO and XPO _initially_ but when  all methods are provided one or two more iterations DPO and XPO catch up? Does SE-POPO degrade faster at higher iteration counts? Whether SE-POPO maintain a consistent and durable lead at all budgets is not clear.
    3. If there is no way to connect the actual goal of the proposed theoretical improvement and how the method performs in the evaluation, then it is generally unclear whether the _reason_ for why SE-POPO is performing better is indeed because of how the scaling of the regret bound is modified. SE-POPO could be better, but not necessarily having anything to do with the $R_{max}$ problem. (see related question)

---

> ### Author Rebuttal · Authors · 2025-07-28
>
> **Summary**: We sincerely thank the reviewer for the thorough, thoughtful, and constructive feedback. The reviewer raises two main concerns:
>
> (1) the lack of explanation in the main text for certain algorithmic design choices, and
>
> (2) the absence of direct empirical evidence demonstrating that the exponential reward scaling issue occurs in practice—and that our algorithm outperforms others because it addresses this problem (i.e., a proof of causality).
>
> Both are excellent points.
>
> Regarding (1): Unlike many works in this space that rely primarily on intuition to justify their algorithms, our method is theoretically grounded—meaning theorems and proofs explicitly explain why the algorithm works. We have been deliberately cautious about including high-level intuitions or informal sketches in the main text, as such explanations often risk oversimplifying or misrepresenting the actual mechanics and can lead to misunderstandings.
>
> Regarding (2): We agree this is an important observation. In response, we have followed your suggestions to:
>
> (a) measure both the average and peak reward gaps during training for our algorithm and the baselines, and
>
> (b) add new experiments using different base models.
>
> Details are provided below.
>
> **Q1: How to interpret Eq. 9 as encouraging exploration?**
>
> **R1:** The high-level intuition behind Eq. 9 closely mirrors that of prior works on optimistic RLHF exploration [1, 2, 3]. These methods aim to minimize reward regret via a surrogate loss of the form:
>
> $\mathbb{E}_{x, y^\star, y \sim \pi(\cdot|x)}[r(x, y^\star) - r(x, y)]$
>
> and incorporate an exploration bonus like:
>
> $\mathbb{E}_{x, y \sim \pi(\cdot|x)}[r(x, y)]$
>
> to encourage visits to uncertain or under-explored regions.
>
> Our algorithm optimizes the preference regret instead of reward regret. Thus, we replace the reward-based bonus with a preference-based one—i.e., substituting $r(x, y)$ with $\mathbb{P}\_r(y \succ y' \mid x)$. Eq. 9 encourages the current policy $\pi$ to produce responses that are preferred over those sampled from $\pi{\text{sam}}$, effectively pushing $\pi$ away from $\pi_{\text{sam}}$ and thus promoting exploration. An MLE-based truncation is then added to prevent over-exploration.
>
>
>
> **Q2: How optimizing Eq. 9 minimizes Eq. 8?**
>
> **R2**: This relationship is formally established in Theorem 3.4. We offer a brief summary of the key logical steps in the proof below.
>
> The preference regret in Eq. 8 can be decomposed into three terms each to be bounded separately.
>
> 1. **Term 1**: $\sum_{t=1}^T \left[G(r^\star, \pi_\beta^\star) - G(r_t, \pi_t)\right]$.
>
>    Optimizing Eq. 9 yields
>    $
>    G(r^\star, \pi_\beta^\star) - G(r_t, \pi_t) \le \frac{1}{\alpha} \left[\ell(r^\star, \mathcal{D}\_{t-1}) - \ell(r_t, \mathcal{D}\_{t-1})\right].
>    $
>    Notice that $r^\star = \arg\min_r \mathbb{E}_{\mathcal{D}}[\ell(r, \mathcal{D})]$ for all $\mathcal{D}$.
>    In this case, if $r_t$ deviates too much from $r^\star$, this component can become negative, thereby canceling out the positive terms in the latter two components.
>
> 2. **Term 2**: $\sum_{t=1}^T \left[G(r_t, \pi_t) - G(r^\star, \pi_t)\right]$.
>
>    This term represents the prediction error, which can be bounded by the estimation errors
>    $\sum_{s=1}^t (r^\star(x,y) - r_t(x,y))^2$.
>
>
> 3. **Term 3**: $\sum_{t=1}^T \left[G(r^\star, \pi^\star) - G(r^\star, \pi^\star_\beta)\right]$.
>
>    This component is bounded directly by the KL divergence between $\pi^\star$ and $\pi_{\text{ref}}$.
>
> Summing the three terms yields a bound on preference regret. This proves that optimizing Eq. 9 indeed reduces Eq. 8.
>
> **Q3: Why can we drop the bonus term from Eq.11 to Eq.12?**
>
> **R3**: Intuitively, Both Eq. 11 and Eq. 12 aim to move the policy $\pi$ away from the sampler $\pi_{\text{sam}}$, albeit via different approximations.
>
> Eq. 11, maximizes the gap of the implicit reward between $y$ sampled from the current policy $\pi$ and $y'$ sampled from the sampler policy $\pi_{\text{sam}}$.
>
> Eq. 12 minimizes $\pi(y' \mid x)$ for $y'$ sampled from $\pi_{\text{sam}}$.
>
> Although simplified, both objectives promote exploration by discouraging the current policy from collapsing toward the sampler.
> The choice to adopt Eq. 12 is due to practical constraints: online sampling from $\pi$ is prohibitively slow for LLM finetuning. Removing this dependency allows implementation in an iterative RLHF pipeline without requiring high-throughput online inference. We intend to explore the full formulation in Eq. 11 as LLM serving becomes more efficient.
>
> ---
> ---
>
> Before addressing the reviewers’ comments on the experiments, we want to point out an observation: the detrimental effect of the exponential dependence on $R_{\text{max}}$ has likely already been observed empirically in prior work. Notably, in previous RLHF exploration studies [1,2,3,4], **all theoretical analyses** are based on using $(\pi_t, \pi_{\text{ref}})$ as the sampler.
>
> However, in practice, these works consistently use $(\pi_t, \pi_t)$ during experiments—despite this choice being **unsupported by their theory**. In light of our results, we believe this deviation is driven by an empirical observation: when the sampler uses $(\pi_t, \pi_{\text{ref}})$, the large reward gap significantly impairs performance. To mitigate this, prior studies often update $\pi_{\text{ref}}$ iteratively, even though this is not part of the theoretical formulation.
>
> In contrast, our work provides the **first formal theoretical justification** for why using $(\pi_t, \pi_{\text{ref}})$ is suboptimal, and why iteratively updating $\pi_{\text{ref}}$ is beneficial. By aligning theory with practice, our algorithm achieves **consistency between the theoretical and the implemented version**, which we believe is essential for principled LLM alignment algorithm design.
>
> **Q4: Does the empirical results support the theory?**
>
> **R4**: We acknowledge that Table 1 alone may not isolate the effect of reward gap control. However, our ablation study (Fig. 1) aims to isolate the effect of reward gap scaling.
>
> We compare samplers $(\pi_t, \pi_t)$ vs. $(\pi_t, \pi_{\text{ref}})$ and observe the following:
>
> (1) Under $(\pi_t, \pi_t)$, reward and win rate continue improving in iteration 3.
>
> (2) Under $(\pi_t, \pi_{\text{ref}})$, reward saturates much earlier—due to large reward gaps making preference labels uninformative (as shown below).
>
> (3) Following the reviewer's suggestion, we also report the reward gap statistics for the training data at Iteration $t = 2, 3$ (when saturation happens):
>
> | Sampler                     | Gap Mean | Gap Max | Ratio (Gap > 3) |
> | --------------------------- | -------- | ------- | --------------- |
> | $(\pi_t, \pi_t)$            | 2.33     | 23.6    | 28%            |
> | $(\pi_t, \pi_{\text{ref}})$ | 3.84     | 28.4    | 49%            |
>
> | Sampler                     | Gap Mean | Gap Max | Ratio (Gap > 3) |
> | --------------------------- | -------- | ------- | --------------- |
> | $(\pi_t, \pi_t)$            | 2.90     | 26.0    | 36%            |
> | $(\pi_t, \pi_{\text{ref}})$ | 7.14     | 33.9    | 71%            |
>
> The results show that updating $\pi_{\text{ref}}$ iteratively indeed reduces the reward gap efficiently, which in turn allows for continuous policy improvement. This aligns with our theory.
>
> In Appendix K.3, we further show that XPO using the theoretical sampler $(\pi_t, \pi_{\text{ref}})$ performs significantly worse than both SE-POPO and XPO with the $(\pi_t, \pi_t)$ sampler.
>
> **Q5:  Additional Experiments**.
>
> **R5**:  We added an experiment using Qwen2.5-7B-Instruct as the base model.  The results can be found in the rebuttal to Reviewer xH4p. After switching the base model, our algorithm is still generally able to outperform DPO and XPO. This further validates the effectiveness of our algorithm.
>
> ---
> ---
>
> **Q6: Definition of $J(r, \pi)$.**
>
> **R6**:     $J(r, \pi)$ is defined in Eq. 2, that is,
>     $J(r, \pi) = \mathbb{E}\_{x\sim \rho, y\sim \pi(\cdot|x)}\left[  r(x,y) - \beta\log \frac{\pi(y|x)}{\pi_{\text{ref}}(y|x)} \right].$
>
> **Q7: Is the preference based regret in Eq. 8 a novel definition in this work?**
>
> **R7**: Yes it is indeed a new definition, to the best of our knowledge. We will style it as a formal definition in the revision.
>
> **Q9: Canonical form of DPO.**
>
> **R9**: Eq. 5 is actually the canonical DPO, where we we consider an offline dataset $\mathcal{D} = (x_n, y_n^w, y_n^l)_{n=1,\dots ,N}$. We will clarify this in the revised version.
>
>
> **Q10: Format Lemma 3.1 on its own line and provide a proof sketch.*
>
> **R10**: Yes, we agree that Lemma 3.1 should be given more context. We did not include this in the original submission mainly due to page limitations.
> Given that the revised version allows for one additional page, we will rewrite the statement of Lemma 3.1 in a single line and provide a proof sketch in the main text.
>
>
> **References**:
>
> [1] Xie et al. Exploratory preference optimization: Harnessing implicit Q*-approximation for sample-efficient. ICLR 2025.
>
> [2] Cen et al. Value-incentivized preference optimization: A unified approach to online and offline RLHF. ICLR 2025.
>
> [3] Zhang et al.  Self-exploring language models: Active preference elicitation for online alignment. TMLR 2025.
>
> [4] Xiong et al. Iterative preference learning from human feedback: Bridging theory and practice for RLHF under KL-constraint. ICML 2024.
>
> ---
>
> We hope the clarifications and updates we’ve provided help to address your concerns. We would appreciate it if you could consider updating the score. Thank you again for your thoughtful review!

---

> ### Comment · Reviewer_rqsZ · 2025-08-05
> **Response**
>
> The authors thorough response and engagement with the points made in the original review are appreciated. In particular, the authors clear identification of what the reviewer's primary concern was, and why it is important was a non-trivial task given the verbosity of the review. Kudos.
>
> ### Comments on selected responses
>
> R2: The reviewer appreciates the author's obliging the request and simply notes that the distance between Eq's 8,9 and Thm 3.4 (and of course the proof in appendix) makes this relationship unclear. A glue sentence can be added that alludes to how this relationship will be formalized somewhere near top of pg 5 where it is first implied.
>
> R3: The clarification is appreciated. The remark still stands that online RL is now commonplace in LLM posttraining (eg. GRPO) even at modest academic compute budgets because the simultaneous training+rollout infrastructure has matured rapidly. Therefore it is not clear that these results under approximation to Eq. 12 are likely to stay as relevant as an analysis based on Eq. 11.
>
> Unnumbered:
> > in practice, these works consistently use $(\pi_t, \pi_t)$ during experiments—despite this choice being unsupported by their theory
>
> The point being raised here is a valuable insight! Is this mentioned anywhere in the manuscript? The connection between a practical fix deployed in prior work and the principled derivation of a solution that embodies it is an extremely useful type of result (in this reviewer's opinion) and does not detract novelty in any way.
> Perhaps this can be incorporated at L142 as this appears to be the moment where the motivating connection makes the most sense in this paper as submitted.
>
> **R4**: This is quite informative, and appears to be concrete evidence that the theoretically grounded intervention both causes an improvement in performance, but that this key quantity targeted by the intervention also improves. This is an important result, and helps establish a evidentiary basis for future work to pursue alternate/complementary improvements by directly targeting the reward gap in other ways.
>
> In general the reviewer appreciates all other responses and hopes that the authors can commit to working in these additional clarifications as space and taste permit.
>
> ### Verdict
>
> The thorough rebuttal has increased the reviewer's understanding and confidence in the work.
> The manuscript will benefit from the additional empirical results provided during the rebuttal as well as the enumerated ways in which each of the theoretical sections are motivated and tied together. In this particular case, it is quite unfortunate that there is no allowed updating of the PDF to seen the work in updated form since the issues essentially hinge on careful presentation and explication.
>
> That said, the reviewer is happy to bump up their score up to the side of weak acceptance.

---

> > ### Author Response · Authors · 2025-08-05
> >
> > Thank you for taking the time to re-evaluate our work and for your thoughtful follow-up! We will make sure to incorporate all the comments above into the revised version of the paper.
> >
> >
> > > The remark still stands that online RL is now commonplace in LLM posttraining (eg. GRPO) even at modest academic compute budgets because the simultaneous training+rollout infrastructure has matured rapidly.
> >
> > We acknowledge this point. For us, the main overhead does not lie in computation but in GPU memory: in typical GRPO tasks, only two LLMs—$\pi_t$ and $\pi_{\text{ref}}$—need to be maintained. However, in our algorithm, we simultaneously hold four LLMs in memory: $\pi_t$, $\pi_{\text{ref}}$, $\pi_{\text{sam}}$, and a reward model. This forces us to reduce the number of rollouts in order to parallelize the models across multiple GPUs, which significantly decreases inference efficiency. We will explore the online pipeline in our future work.
> >
> >
> > > ... Is this mentioned anywhere in the manuscript?...
> >
> > We mentioned this point a bit in Lines 284 to 288. We will incorporate a clear clarification on the differences between the theoretical and practical implementation of prior works in the revised version.

---

### Official Review · Reviewer_xH4p · 2025-07-02

**Clarity:** 3
**Significance:** 3
**Originality:** 2
**Rating:** 4
**Confidence:** 4

**Summary:**

The authors theoretically analyze the issue that existing online RLHF algorithms suffer from sample complexity that grows exponentially with the reward range, leading to inefficiency and poor exploration in scenarios with extreme preferences (e.g., tasks with objectively correct answers). To address this, they propose an algorithm similar to *PO, named SE-POPO, which resolves the aforementioned exploration problem. The authors further provide a theoretical proof that the sample complexity of their method scales polynomially with the reward range, overcoming the exponential limitation of existing algorithms.

**Questions:**

Please address all the issues mentioned in the Weaknesses section as thoroughly as possible.

**Ethical Concerns:**

["NO or VERY MINOR ethics concerns only"]

**Final Justification:**

My concerns have mostly been addressed.

**Limitations:**

yes

**Quality:**

4

**Strengths And Weaknesses:**

### Strengths
- The paper presents its arguments clearly, analyzing and addressing the problem of exponential sample complexity.
- The proposed method is engineering-friendly, simple to implement, and can be directly integrated into existing DPO frameworks.
- Experimental results demonstrate the effectiveness of the proposed approach.

### Weaknesses

- The experiments appear to have been conducted only on Llama-3.1-8B. Could the authors evaluate their method on additional models, such as the Qwen series, to further validate its effectiveness?
- Could the authors provide additional statistics on computational overhead? Iteratively updating the sampler may increase the computational resources required during training.
- The experiments use Llama-3.1-8B-SFT and compare the results with Llama-3.1-8B-Instruct to demonstrate the method’s effectiveness. Could the authors also conduct experiments directly on Llama-3.1-8B-Instruct to further substantiate their claims?

### Additional Question

- Could the authors further explain the meaning of Equation (6)? My understanding is that the training data for the reward model in the next stage should be sampled from responses that the latest policy receives high scores from the previous round’s reward model, in order to prevent reward hacking. I would appreciate a more detailed explanation to clarify this point.
- Could you please explain how Llama-3.1-SFT was trained? Was the RLHFlow-Ultrafeedback dataset used as the training set? However, it seems that this dataset is pairwise in nature.

---

> ### Author Rebuttal · Authors · 2025-07-30
>
> **Q1**: The experiments appear to have been conducted only on Llama-3.1-8B. Could the authors evaluate their method on additional models, such as the Qwen series, to further validate its effectiveness?
>
> **R1**: Sure! Here we added an experiment using Qwen2.5-7B-Instruct as the base model.  The results are below.
>
>
>
> | Model         | IID Data |           | Alphca Data |          |
> | ------------- | -------- | --------- | ----------- | -------- |
> |               | WR       | AvgV      | WR          | AvgV     |
> | Base model    |          | -2.86     |             | -3.39    |
> | DPO-iter1     | 61.7     | -1.93     | 64.2        | -2.20    |
> | DPO-iter2     | 67.1     | -1.26     | 78.3        | -0.81    |
> | DPO-iter3     | 71.7     | -0.55     | 84.4        | 0.66     |
> | XPO-iter1     | 61.8     | -1.92     | 69.5        | -1.80    |
> | XPO-iter2     | 68.5     | -1.14     | 81.2        | -0.27    |
> | XPO-iter3     | 73.5     | **-0.19** | 86.0        | 0.97     |
> | SE-POPO-iter1 | 62.5     | -1.86     | 68.4        | -1.72    |
> | SE-POPO-iter2 | 69.2     | -1.09     | 81.7        | -0.10    |
> | SE-POPO-iter3 | **74.1** | -0.21     | **86.6**    | **1.18** |
>
> As shown, SE-POPO consistently outperforms both DPO and XPO, even after switching to a different base model. This further validates the robustness and effectiveness of our method.
>
> **Q2**: Could the authors provide additional statistics on computational overhead? Iteratively updating the sampler may increase the computational resources required during training.
>
> **R2**:
> In our experiments, we adopt an **iterative pipeline**, where the sampler is updated only **between iterations** (i.e., three times in total). This incurs **no additional overhead** beyond what is already required in baseline methods.
>
> In a fully online setting, modern RLHF frameworks such as** VERL** support efficient sampler updates. Reloading the LLM’s parameters is inexpensive compared to the overall cost of inference and gradient updates. Thus, we do not observe any significant computational overhead from sampler updates in either theoretical or practical implementations.
>
> **Q3**: The experiments use Llama-3.1-8B-SFT and compare the results with Llama-3.1-8B-Instruct to demonstrate the method’s effectiveness. Could the authors also conduct experiments directly on Llama-3.1-8B-Instruct to further substantiate their claims?
>
> **R3**: Due to time and resource constraints, we were unable to run additional experiments with LLaMA-3.1-8B-Instruct as the base model. However, in R1, we present results using **Qwen2.5-7B-Instruct** as the base model. SE-POPO still achieves significant performance gains, suggesting that our algorithm is effective even when starting from an instruct-tuned base model.
>
> **Q4**: Could the authors further explain the meaning of Equation (6)? My understanding is that the training data for the reward model in the next stage should be sampled from responses that the latest policy receives high scores from the previous round’s reward model, in order to prevent reward hacking. I would appreciate a more detailed explanation to clarify this point.
>
> **R4**: Great question—here is an intuitive explanation.
>
> Equation (6) is motivated by the **optimism principle* from classical bandit and RL algorithms, such as UCB. In the bandit setting, UCB selects actions by solving:
>
> $
> \pi_{t+1} = \arg\max_{\pi} \left( \bar{r}(y) + \alpha \cdot b(y) \right),
> $
>
> where $\bar{r}(y)$ is the empirical mean and $b(y)$ is an optimism bonus.
>
> For LLMs, directly applying UCB is infeasible because the response space is unstructured and high-dimensional. Instead, we reformulate the bonus via a constrained optimization:
>
> $
> \arg\max_{\pi} \max_{r \in \mathcal{R}(\mathcal{D}_t)} J(r, \pi),
> $
>
> where $\mathcal{R}(\mathcal{D}_t)$ denotes a neighborhood around the MLE reward $\bar{r}$.
>
> Relaxing the constraint leads to the regularized form:
>
> $
> \arg\max_{\pi} \max_{r} \left( J(r, \pi) - \frac{1}{\alpha} \ell(r, \mathcal{D}_t) \right),
> $
>
> which is essentially Equation (6). Rather than filtering samples to prevent reward hacking, this approach encourages policies that perform well under optimistic but plausible reward functions.
>
>
>
> **Q5**: Could you please explain how Llama-3.1-SFT was trained? Was the RLHFlow-Ultrafeedback dataset used as the training set? However, it seems that this dataset is pairwise in nature.
>
> **R5**: The base model used in our experiments was **not trained by us**. As noted in Appendix K.1, we use the publicly released model **RLHFlow/LLaMA3-SFT** from Huggingface.
>
> Training details for this model can be found in Section B.3 of Dong et al. [1], and confirm that it was trained using supervised instruction data—not directly on pairwise preference data like RLHFlow-Ultrafeedback.
>
>
>
> **References**:
>
> [1] Dong et al. "Rlhf workflow: From reward modeling to online rlhf." arXiv preprint arXiv:2405.07863 (2024).
>
> ---
>
> We hope the clarifications and updates we’ve provided help to strengthen your confidence in the contribution. Thank you again for your thoughtful review.

---

> > ### Comment · Reviewer_xH4p · 2025-08-04
> > **Response to authors**
> >
> > Thank you for your response. My concerns have been almost addressed. Please make sure the new experimental results are incorporated into the latest version of your manuscript.

---

> > > ### Author Response · Authors · 2025-08-05
> > >
> > > Thank you for taking the time to read our rebuttal! We will make sure to incorporate the new experimental results into the revised version of the paper.

---

### Official Review · Reviewer_NcxC · 2025-07-03

**Clarity:** 3
**Significance:** 2
**Originality:** 3
**Rating:** 4
**Confidence:** 3

**Summary:**

This paper addresses the problem of sample inefficiency in online Reinforcement Learning from Human Feedback (RLHF) in theory. The authors identify a key limitation in existing theoretical online RLHF algorithms: their sample complexity scales exponentially with the range of the underlying reward function ($R_{\max}$). To overcome this, the paper introduces a new algorithm called SE-POPO. Instead of using a reward-based exploration bonus like prior work, SE-POPO uses a preference-based bonus. This directly incentivizes the policy to find responses that are preferred over a given sampler policy, which is shown to be more efficient when the reward scale is large. Besides, SE-POPO iteratively calls a subroutine POPO that uses the preference-based exploration. The output policy from each iteration is then used as the sampler for the subsequent iteration. Theoretically, the paper demonstrates that SE-POPO is the first algorithm to achieve a sample complexity that scales polynomially with the reward range. Empirically, the authors conduct extensive experiments on LLM alignment tasks, comparing SE-POPO against state-of-the-art passive (DPO) and active (XPO) exploration baselines. The results show that SE-POPO consistently outperforms these baselines across various settings.

**Questions:**

1. In the POPO objective (Equation 9), the paper introduces an indicator function I(r, D_t) to truncate the exploration bonus. However, the paper mentions in Section 3.3 (lines 202-203) that this truncation is "rarely active and can therefore be omitted" in the practical implementation. Is this truncation term primarily a theoretical tool needed to secure the proofs, or are there practical scenarios where it would be crucial?
2. SE-POPO iteratively updates the sampler $\pi_{sam}$ by running the POPO subroutine for T iterations. In experiments (Section 4 and Appendix K.1), the paper uses a 3-iteration implementation. Could the authors clarify the relationship between the number of "intervals" K in Algorithm 1 and the "iterations" in the experiments?

**Ethical Concerns:**

["NO or VERY MINOR ethics concerns only"]

**Limitations:**

Yes.

**Paper Formatting Concerns:**

None.

**Quality:**

3

**Strengths And Weaknesses:**

Strengths:

1. Significance: The paper tackles a fundamental problem in the theory of online RLHF. The exponential dependency on the $R_{\max}$ is a major drawback of the prior theoretical works. By being the first to propose an algorithm with polynomial scaling, this work answers an open question from [Xie et al., 2024] and significantly advances the state of the art. The ideas presented are likely to be highly influential for future research and practice in LLM alignment.
2. Originality: The technical contributions are highly original. This paper proposes the Preference-based Exploration, which shifts from a standard reward-based optimism bonus to a preference-based one. It directly targets the source of information (preferences) rather than an inferred, and potentially poorly scaled, proxy (rewards). This is a clever and effective idea. Besides, the mechanism of iteratively updating the sampler policy to "bootstrap" performance is an elegant solution to avoid comparing against a weak or fixed reference policy. This gradual improvement strategy is what ultimately breaks the exponential dependency, as shown in Theorem 3.5.

Weaknesses:

1. Significance: The entire theoretical framework are built upon the bandit formulation, which deviates from the token-level MDP used in practice. Besides, this paper assumes the BT preference model, which may not capture the complex human preferences. The above two gaps may restrict the impact of this paper.
2. Quality: As discussed in experiments, the proposed method SE-POPO introduces a bias towards generating longer responses. As such, a natural question is whether the better empirical performance of SE-POPO is mainly due to this length bias. This paper may benefits from more empirical investigations on the length bias.

---

> ### Author Rebuttal · Authors · 2025-07-28
>
> **Q1**:     The entire theoretical framework are built upon the bandit formulation, which deviates from the token-level MDP used in practice.
>
> **R1**:    Thank you for the question.
> For LLMs, given a response sequence $y = {y_1, \ldots, y_n}$, the sequence likelihood decomposes as
>
> $\pi(y|x) = \prod_k \pi(y_k \mid x, y_{1:k-1}).$
>
> As such, our bandit-level formulation can be naturally extended to the token-level MDP by applying our method autoregressively at each token step. This approach is standard in the literature and is adopted in prior work (e.g., see [1], Appendix C.2). We will include a discussion of this connection in the revised version.
>
> **Q2**:      This paper assumes the BT preference model, which may not capture the complex human preferences.
>
> **R2**:     This is a valid point.
> We note that the $\exp(R_{\max})$ issue addressed in our work is specific to reward-based preference models—i.e., where preferences are a deterministic function of an underlying reward difference. For general models that do not assume a latent reward (e.g., direct preference learning), the notion of $R_{\max}$ does not apply directly.
>
> However, our framework extends naturally to any reward-based preference model of the form:
>
> $\mathbb{P}(y \succ y' \mid x) = f(r(x, y) - r(x, y'))$,
>
> where $f$ is any smooth, monotonic link function. In such cases, we can replace the logistic function $\sigma$ in our analysis with $f$, and our algorithm and regret bounds continue to hold under mild regularity assumptions. We will clarify this generalization in the revised version.
>
>
> **Q3**:      The proposed method SE-POPO introduces a bias towards generating longer responses. As such, a natural question is whether the better empirical performance of SE-POPO is mainly due to this length bias.
>
> **R3**:       We believe the performance improvement of SE-POPO is not primarily due to response length, for two reasons:
>
> 1. The results from AlpacaEval are based on **length-controlled** win rates, which are explicitly adjusted to correct for any length bias.
>
> 2. By iteration 3, the average response lengths of XPO and SE-POPO are nearly identical.
>
> Thus, the observed performance gains over DPO and XPO cannot be attributed to response length and reflect genuine improvements in policy quality.
>
>
> **Q4**:     Is this truncation term primarily a theoretical tool needed to secure the proofs?
>
> **R4**:     Yes, that is correct.
> The truncation condition is used to bound Term 2 in the proof of Theorem 3.4 (specifically Line 636), ensuring the MLE-based reward does not deviate too far from the learned reward.
>
> In practice, however, we set the exploration coefficient $\alpha$ to be much smaller than the theoretical requirement of $O(1/\sqrt{T})$. This keeps the deviation between $r_t$ and $\bar{r}$ small, so the condition
>
> $\ell(r_t, \mathcal{D}_t) - \ell(\bar{r}, \mathcal{D}_t) < \gamma$
>
> almost always holds.
>
> For this reason, and to reduce computational overhead, we omit the truncation term in our experimental implementation. We will clarify this in the paper.
>
> **Q5**: Could the authors clarify the relationship between the number of "intervals" K in Algorithm 1 and the "iterations" in the experiments?
>
> **R5**:     The "iterations" in our experiments correspond directly to the $K$ intervals in Algorithm 1. Due to computational constraints, we adopt an iterative pipeline rather than a fully online one. This setup is consistent with prior work [1,2,3].
>
> Concretely, in each iteration, we:
>
> 1. Generate a batch of preference data,
>
> 2. Run our algorithm using the collected data,
>
> 3. Update the sampler $\pi_{\text{sam}}$.
>
> This iterative process implements the interval-wise updates described in SE-POPO. Full implementation details are provided in Appendix K.1 (Page 28).
>
>
> **References**:
>
> [1] Xie, et. al. Exploratory preference optimization: Harnessing implicit Q*-approximation for sample-efficient. ICLR 2025.
>
> [2] Cen, et. al. Value-incentivized preference optimization: A unified approach to online and offline RLHF. ICLR 2025.
>
> [3] Zhang, et. al.  Self-exploring language models: Active preference elicitation for online alignment. TMLR 2025.
>
>
>
> ---
>
> We hope these clarifications address your concerns and help strengthen your confidence in the paper. Thank you again for your thoughtful review and feedback.

---

> > ### Comment · Reviewer_NcxC · 2025-08-09
> >
> > Thanks for the authors' detailed responses. I have no more questions and will keep my original positive score.

---

### Official Review · Reviewer_hnSu · 2025-07-03

**Clarity:** 3
**Significance:** 3
**Originality:** 3
**Rating:** 5
**Confidence:** 3

**Summary:**

This paper introduces SE-POPO, an online Reinforcement Learning from Human Feedback (RLHF) algorithm that avoids the exponential dependence on the reward scale ($R_\max$) that appears in various previous theoretical works. Existing RLHF approaches, even with active exploration, suffer from sample complexity that scales exponentially with $R_\max$ due to the nonlinear nature of human preference modeling via the Bradley–Terry model. SE-POPO addresses this by introducing a self-updated sampler strategy that is key to achieving polynomial sample complexity in the leading term. Theoretical analysis establishes regret bounds without exponential scaling, and empirical results show SE-POPO outperforms previous baselines (DPO, XPO) on alignment benchmarks and public datasets.

**Questions:**

1. In Line 224, $\phi(x,y,y’)$ should be $\phi(x,y)$ for all $x,y$. The same problem also exists throughout the proof of Theorem 3.4 (Appendix D). This casts concerns to the overall correctness of the proof.
2. In the last equation in Page 17, how is the second to last inequality derived? Does it make sense to directly bound the derivative of sigmoid by constant here? since here the goal is to bound the preference difference by reward difference, the direction that does not incur exponential factors.
3. Typos in Equation (14).
4. From my understanding of the proof, the exponential dependency still exists for the logarithm term, despite that it is a non-leading term (in terms of $N$). However, this means that the problem still suffers from a burn-in cost that is exponential in the scaling of the reward. I suggest presenting this explicitly in the main results, as opposed to hiding all the non-leading terms in terms of $N$.
5. Is it possible to avoid such exponential dependency for offline RLHF setups? where only fixed preference data are provided.

**Ethical Concerns:**

["NO or VERY MINOR ethics concerns only"]

**Final Justification:**

The clarity issue can be resolved, and the contribution of the paper is significant to the community. Thus the reviewer is recommending acceptance of the paper.

**Limitations:**

Please see the Questions part.

**Quality:**

2

**Strengths And Weaknesses:**

**Strengths:**
1. The theoretical results are significant and interesting. The analysis tailored for the sampler updating mechanism could be of independent interest for future research.
2. Experimental results demonstrate the empirical effectiveness of the theoretical algorithm to some extent.

**Weaknesses:**
1. The POPO subroutine is relatively direct given some prior works on provably sample-efficient online RLHF with reward bonus to incentivize exploration.
2. Some unclarity issues, please see the Question part.

---

> ### Author Rebuttal · Authors · 2025-07-28
>
> We thank the reviewer for giving our paper a thorough and careful read, including the appendix! Below we address some of the reviewer's questions.
>
> **Q1: The POPO subroutine is relatively direct given some prior works on provably sample-efficient online RLHF with reward bonus to incentivize exploration.**
>
> **R1**: We appreciate the reviewer’s observation and would like to clarify an important distinction. The fact that the POPO subroutine is simple and implementation-friendly is a deliberate strength of our design—not a limitation. However, while the algorithm is indeed straightforward to implement, the underlying analysis that enables this simplicity is far from trivial.
>
> In particular, our proof introduces several new techniques and leverages specific properties of the logistic function that have not been explored in prior work. For example, in bounding Term 2 (Lines 631–657), we carefully analyze the gradient behavior of the logistic function (see Lines 631, 653, and 656) to eliminate the exponential dependence on $R_{\max}$. Avoiding this $\exp(R_{\max})$ factor is non-trivial and requires technical innovations beyond what existing approaches provide.
>
> These insights are central to our contribution and reflect a fundamental difference between our method and prior work: POPO achieves both theoretical tightness and practical usability without compromising either.
>
>
> **Q2: In Line 224, $\phi(x,y,y')$ should be $\phi(x,y)$ for all $x,y$.**
>
> **R2**: Thank you for catching this!
> You are absolutely right that this could have been stated more clearly. In our formulation, we define $\phi(x, y, y') := \phi(x, y) - \phi(x, y')$. This implies that
>
> $r(x, y) - r(x, y') = \langle \theta, \phi(x, y, y') \rangle$,
>
> which is the quantity used throughout the proof in Appendix D. Therefore, the derivation remains correct as written.
>
> We will include the explicit definition of $\phi(x, y, y')$ in the revised version to avoid confusion.
>
>
> **Q3: In the last equation in Page 17, how is the second to last inequality derived?**
>
> **R3**:   Thank you for the question—let us walk through the key steps in the derivation:
>
> **Second to third inequality:**
> We use the bound
> $\sigma(a) - \sigma(b) \le \min\left( \max_{v \in [-|b|, |a|]} \dot{\sigma}(v) \cdot |a - b|, , 1 \right)$,
> which holds because $\dot{\sigma}(v)$ is symmetric and $\sigma$ is 1-Lipschitz bounded above by 1.
>
> **Third to fourth inequality:**
> We apply the inequality
> $\max_{v \in [-|b|, |a|]} \dot{\sigma}(v) \le \dot{\sigma}(|b|) \cdot \exp(|a - b|)$,
> which follows from the exponential tail behavior of the logistic derivative.
>
> **Fourth to final inequality:**
> We use the inequality
> $\max(\exp(a) b, 1) \le \min(a^2, 1) + \exp(1) b$
> for any $a, b > 0$, which allows us to split the expression into a squared error and a scaled term—both of which are easier to control in the regret bound.
>
> We will make these steps more explicit in the appendix of the revised version to aid readability.
>
> **Q4:     Does it make sense to directly bound the derivative of sigmoid by constant here?**
>
> **R4**:     Unfortunately, this approach does not work.
> If we directly upper-bound the sigmoid derivative by a constant, then Term 2 would take the form
> $\min(|\langle \theta_t - \theta^\star, \phi(x, y, y') \rangle|, 1)$
> rather than
> $\min(|\langle \theta_t - \theta^\star, \phi(x, y, y') \rangle|^2, 1)$.
>
> This linear form is too loose to yield a meaningful bound: in particular, the key inequality used in Line 638 relies on a squared error term. Without the quadratic dependence, the bound would not hold, and Term 2 could dominate the regret, making the overall guarantee vacuous.
>
> **Q5: Typos in Equation (14).**
>
> **R5**:     Thanks for pointing out! It should be $\text{Ref}\_{\text{pref}}(\pi\_{\text{sam}}^k, T)$ instead of $\text{Ref}\_{\text{pref}}(\pi\_{\text{sam}}, T)$.
>     We will update it in the revision.
>
> **Q6:     From my understanding of the proof, the exponential dependency still exists for the logarithm term, despite that it is a non-leading term (in terms of $N$). However, this means that the problem still suffers from a burn-in cost that is exponential in the scaling of the reward. I suggest presenting this explicitly in the main results, as opposed to hiding all the non-leading terms in terms of $N$.**
>
> **R6**:     Thank you for the insightful comment—you are absolutely right.
> While our algorithm eliminates the exponential dependence in the leading sample complexity term, it still incurs an $\exp(R_{\text{max}})$ burn-in cost in the lower-order term. This is consistent with known lower bounds in the logistic bandit literature (e.g., [1]) and is believed to be unimprovable.
>
> We agree this should be made explicit. In the revised version, we will update Theorems 3.4 and 3.5 to clearly reflect this exponential burn-in term and will add a corresponding remark in the main text to highlight this limitation.
>
> [1] Faury, Louis, Marc Abeille, Clément Calauzènes, and Olivier Fercoq. "Improved optimistic algorithms for logistic bandits." In International Conference on Machine Learning, pp. 3052-3060. PMLR, 2020.
>
> **Q7:     Is it possible to avoid such exponential dependency for offline RLHF setups? where only fixed preference data are provided.**
>
> **R7**: Intuitively, this is not possible without assuming a much stronger coverage condition on the offline dataset.
>
> Consider a simple illustrative example with three responses: $a$, $b$, and $c$, where $r(a) > r(b) \gg r(c)$. Suppose the offline dataset only includes preference pairs $(a, c)$ and $(b, c)$. In this case, both $a$ and $b$ will always be labeled as the preferred response, since they both significantly outperform $c$. However, the dataset provides no information to distinguish between $a$ and $b$, as the pair $(a, b)$ is never observed.
>
> To resolve this ambiguity, one would need to infer the relative quality of $a$ and $b$ indirectly through their comparisons to $c$. But because $r(b) - r(c)$ is large, accurately estimating the preference probabilities in this transitive chain requires at least $O(\exp(r(b) - r(c)))$ samples—i.e., exponential in $R_{\max}$. This reflects a known hardness in preference-based inference under limited pairwise supervision.
>
> Therefore, avoiding the exponential dependency in the offline setting would require strong assumptions—such as full pairwise coverage of all responses—which are rarely met in practice.
>
> ---
>
> We hope the clarifications and updates we’ve provided help to strengthen your confidence in the contribution. Thank you again for your thoughtful review.

---

> > ### Comment · Reviewer_hnSu · 2025-08-03
> >
> > Thank you for your detailed response to all my concerns. Please make sure the clarity issues are resolved in revision. I have no more questions and I have increased my score accordingly.

---

> > > ### Author Response · Authors · 2025-08-05
> > >
> > > Thank you for taking the time to read our rebuttal and increasing the score! We will make sure to incorporate all the things above into the revised version of the paper.

---

### Decision · Program_Chairs · 2025-09-17

**Decision:**

Accept (poster)

**Comment:**

This paper tries to address the sample inefficiency in RLHF from a theoretical aspect. All reviewers agreed that the results are interesting and promising and admired the novelty of this paper.

The reviewers also raised some concerns, such as missing experiments,  computational overhead, and questions about experimental setup. After the rebuttal, all reviewers reached an agreement and recommend to accept it.

I'm happy to recommend to accept this submission, while the authors should incorporate the suggestions from all reviewers and update the paper in the camera-ready version.